

# Quantification of seasonal variabilities in groundwater discharge in an extensive irrigation watershed using H, O, and Sr isotopes

Takeo Yoshida[1], Takanori Nakano[2,3], Ki-cheol Shin[3], Takeo Tsuchihara[1], Hiroki Minakawa[1], Susumu Miyazu[1], and Tomijiro Kubota[1]

[1]Institute for Rural Engineering, National Agriculture and Food Research Organization, Tsukuba, Japan
[2]Faculty of Science and Engineering, Waseda University, Tokyo, Japan
[3]Research Institute for Humanity and Nature, Kyoto, Japan

**Correspondence:** Takeo Yoshida (takeoys@affrc.go.jp)

**Abstract.**

Numerous studies have quantified stream–groundwater interactions using geochemical or environmental tracers. However, in watersheds where water is extensively used for rice paddy irrigation, uncertainties in estimation remain due to kinetic fractionation of stable isotopes during evaporation from ponded paddies and seasonal variations of the isotopic composition of recharged water. In this study, we used three different methods (streamflow observation, stable isotopes of water, and Sr
5 isotopes) to quantify groundwater discharge to streams in a watershed substantially impacted by rice paddy irrigation in central Japan. We conducted point- and watershed-scale observations of surface water, soil water, groundwater, and ponded water in rice paddies and examined changes in these isotopic compositions. Point-scale observations revealed that Sr isotopes are more appropriate for quantification because the Sr isotopes in groundwater was significantly different from surface water and less variable in time compared to water isotopes. At watershed-scale, isotopic compositions of stream water changed linearly
10 from upstream end to downstream end, suggesting streamflow consisted of two endmembers. We then quantified groundwater discharge to the stream based on measurement of streamflow and surface lateral inflow/outflow during both irrigation and non-irrigation periods. This water balance method yielded large uncertainties in the estimation due to errors in streamflow measurement, while Sr isotopes provided well constrained estimates during both irrigation and non-irrigation periods. The ratios of groundwater to the stream, estimated from Sr isotopes, was in the range 7–86% during the irrigation period and 38–
15 66% during the non-irrigation period. Stable isotopes of water also provided good estimates during the non-irrigation period but underestimated groundwater discharge during the irrigation period due to the ill-defined groundwater end member. The use of Sr isotopes has the potential to aid in quantification of temporal variations in groundwater discharge and to provide important information for water resource managers.

## 1 Introduction

Stream–groundwater interactions take place at various spatial scales. Hyporheic exchange is an interaction of water between the active channel and subsurface flowpaths; the depth of hyporheic exchange is usually in the order of $10^{-2}$ to 1 m beneath the streambed (Harvey and Wagner, 2000). At larger scales, streams interact with adjacent aquifers, with two types of stream-





groundwater interactions: (1) a gaining stream, in which groundwater discharges contribute to streamflow, and (2) a losing stream, which loses water to the aquifer. Numerous studies have quantified stream–groundwater interactions using geochemical or environmental tracers (e.g., McCallum et al., 2012; Banks et al., 2011; Xie et al., 2016). However, the connectedness and degree of interaction can often be altered by agricultural activities, including surface water diversion and groundwater pumping (Tian et al., 2015; McCallum et al., 2013; Yoshioka et al., 2016). Especially in watersheds containing extensive surface irrigation systems for rice paddies, typical of humid regions, interactions can be affected by substantial volumes of irrigation water. Large portions of irrigation water that recharge regional aquifers are later discharged from aquifers to rivers. Discharge from aquifers can play a critical role in low-flow regimes in such streams, and can provide water resources for water users downstream (Yu-Chuan et al., 2016).

The stable isotopes of water, $\delta^2$H and $\delta^{18}$O, have been used to estimate the contribution of irrigation water to regional aquifer recharge (e.g., Tsuchihara et al., 2011; Yoshioka et al., 2016) . However, in watersheds where water is extensively used for rice paddy irrigation, uncertainties in estimation remain due to kinetic fractionation of stable isotopes during evaporation from ponded paddies and seasonal variations of the isotopic composition of recharged water (Tsuchihara et al., 2011). Combinations of several geochemical and environmental tracers may complement the shortcomings of each approach and provide us with new insights about hydrological processes. Based on this idea, Yoshioka et al. (2016) examined recharge of a shallow aquifer in an alluvial fan with multiple tracers to assess the relative contribution of streamwater to aquifers.

Whilst groundwater recharge from rice paddies has been studied, relatively few studies have investigated the contribution of groundwater discharge during irrigation periods. Tsuchihara et al. (2009) used the stable isotopes of water and radon (Rn) as proxies to evaluate discharge from an aquifer to a stream; they compared results during irrigation and non-irrigation periods in watersheds with extensive areas of rice paddies. However, quantification of groundwater discharge in the stream was confounded by the rapid dissipation of Rn. More recently, the existing ratio of stable isotopes of strontium (Sr), $^{87}$Sr/$^{86}$Sr, has been applied in hydrological studies (e.g., Nakano et al., 2008; Shand et al., 2007; Négrel et al., 1999; Petelet-Giraud et al., 2016). In hydrological studies, $^{87}$Sr/$^{86}$Sr has provided valuable information about sources, pathways, and mixing of water (Shand et al., 2007; Négrel et al., 2004) and also about surface water–groundwater interactions in watersheds and groundwater systems (Banks et al., 2011; Land et al., 2000). However, there have been few applications of $^{87}$Sr/$^{86}$Sr to environments substantially impacted by human activities such as irrigation, and the applicability of $^{87}$Sr/$^{86}$Sr as a hydrological tracer in such systems has not been tested.

In this study, we quantified the relative contribution of groundwater discharge to a stream in a watershed containing extensive areas of irrigated paddies in central Japan. We used three different methods, including differential streamflow measurement, and end member analysis based on the stable isotopes of water and Sr. To evaluate the relative contribution of groundwater discharge, we addressed the following questions with respect to determination of the end members: (1) How the isotopic composition of irrigation water changes via surface and subsurface drainage pathways of rice paddies (2) whether groundwater discharges estimated using different methods are consistent with each other, and (3) what the effects of irrigation on temporal changes in stream-groundwater interactions in agricultural watersheds are. To address these questions, we first conducted a point-scale survey in a single rice paddy and then carried out watershed-scale surveys during both irrigation and non-irrigation





periods. Lastly, we quantified the ratio of groundwater to streamflow along the stream. In this paper, we discuss the applicability of each method to elucidate hydrological functions of the watershed.

## 2 Methods

### 2.1 Study watershed and field survey

#### 2.1.1 Study watershed

This study was carried out in the Gogyo River watershed, a typical agricultural watershed on an alluvial fan in central Japan. The Gogyo River is one of the tributaries of the Kinu River, which drains an area of 1761 $\text{km}^2$ (Fig. 1). The Gogyo River originates near the apex of the alluvial fan in the middle reaches of the Kinu River (Fig. 2). Land use in the Kinu River watershed is composed of forest (65%), urbanized areas (12%), and agricultural land (23%). Agricultural land is found in the middle and lower portions of the watershed, and most of it is used for irrigated rice paddies. The growing season for rice starts between late April and early May and ends in September. The climate in the Gogyo River watershed is characterized by hot and wet summers and dry and cold winters. Mean annual precipitation is 1500 mm, 70% of which falls from April through September.

Three major diversion weirs in operation during the growing season divert at most 71 $\text{m}^3\text{s}^{-1}$ of water from the Kinu River. The diverted water is then allocated through irrigation channels and supplied to rice paddies. The Gogyo River originates from spring water of the Kinu River, and like other tributaries on the alluvial fan, it is fed by precipitation, discharge from the shallow aquifer, and irrigation water diverted from the Kinu River. The upper node of the Gogyo River is connected to an irrigation channel, the Ichinohori Channel, which is diverted from the Kinu River. The inflow from the Ichinohori Channel to the Gogyo River is controlled with a gate. During irrigation periods, the gate is opened to let irrigation water flow into the Gogyo River. Opening of the gate is less frequent during the non-irrigation period. The water managers of the area usually open the gate in the middle of April, decreasing openings continuously from September through December, and closing the gate completely from January until the beginning of the next irrigation season.

The Gogyo River then converges with the Kokai River, an adjacent river to the east that drains an area of 1043 $\text{km}^2$ (Fig. 1). Therefore, a fraction of the irrigation water diverted from the Kinu River does not return to the original Kinu River, but instead flows into the adjacent Kokai River. The return flow of irrigation water from the Gogyo River watershed is the primary water resource for the Kokai River watershed because the latter does not include a sufficient mountainous area to support adequate streamflow for downstream farmers.

#### 2.1.2 Field surveys

We carried out surveys across the Gogyo River watershed in June 2016 (during the irrigation period) and October 2016 (during the non-irrigation period). The surveys consisted of (1) measurements of the flow rate in the stream, diversion weir, and drainage



channels along the Gogyo River, (2) sampling of the streamflow along the Gogyo River at approximately 500-m intervals (23 samples), and (3) sampling of groundwater at wells in the watershed (46 locations in Fig. 2).

In addition to these two watershed-scale surveys, we conducted a point-scale survey within the watershed in August 2016 to examine how the stable isotopes of water and $^{87}$Sr/$^{86}$Sr of irrigation water changed within the irrigation and drainage system of

the rice paddies (Fig. 3). More specifically, we investigated changes in the $^{87}$Sr/$^{86}$Sr and water isotopes along two distinct flow paths of the irrigation–drainage system: drainage water from surface pathways and discharge from shallow aquifers. Samples included (1) water from the irrigation channel; (2) water from drainage channels (one sample from channel with soil bottom and two from channels with concrete bottoms); (3) ponding water from rice paddies (one sample from near the inlet and two from near the outlet); (4) water from a spring near the paddy field (hereafter, SAK); and (5) soil water at depths of 0.5 m, 1.0

m, and 1.5 m from the ground surface.

At each sampling site, we collected two bottles of water (50 mL), one for Sr isotopes and the other for stable isotopes of water. Both samples were filtered through 0.20 μm membrane filters. The Sr in the filtrates was separated by cation exchange chromatography on AG50W-X8 200-400 mesh resin (Muromachi Technos Co., Tokyo, Japan) with 2 N HCl as the eluent: the HCl had been purified by evaporating 6 N HCl in 2-l Teflon bottles at low temperature using an infrared heater. $^{87}$Sr/$^{86}$Sr

were determined by means of multicollector double-focusing ICP-MS (Neptune Plus, Thermo Fisher Scientific, Bremen, Germany) with an instrument installed at the Research Institute for Humanity and Nature (Kyoto, Japan). The values of $^{87}$Sr/$^{86}$Sr were normalized to a $^{87}$Sr/$^{86}$Sr of 8.375209. Replicate analyses of the NIST987 standard during this study gave a $^{87}$Sr/$^{86}$Sr of $0.710238 \pm 0.000022$ ($n = 230$). The standard deviation of the $^{87}$Sr/$^{86}$Sr of all samples was less than 0.000015. All measurements were normalized to the $^{87}$Sr/$^{86}$Sr of 0.71025 as recommended by Faure and Mensing (2009). Water isotopes were

analyzed with Piccaro L2140-i at the Institute for Rural Engineering.

## 2.2 Estimation of groundwater discharge to stream

### 2.2.1 Water balance in stream sections

Groundwater discharge can simply be estimated from the difference between streamflow measurements at upstream and downstream locations. We measured the streamflow of the river at approximately 500 m intervals, corresponding to the location of

23 bridges, from Br-1 at the upstream end to Br-23 at the downstream end. We measured streamflow as well as lateral fluxes of surface water at every location with inflow or outflow from the stream. Inflow and outflow of water involved drainage channels and diversion weirs, respectively. There were 30 drainage channels and 22 diversion weirs. We used these data to calculate net surface inflow to the stream in every section defined by the bridges. The volume of net surface inflow was obtained by subtracting the total volume of water diverted from the river from the total volume of river inflow. A positive difference indicated

that the volume of inflow exceeded the volume diverted.

$$S_{\mathrm{net}}(i) = \int S_{\mathrm{in}}(i) - \int S_{\mathrm{out}}(i) \tag{1}$$

where $S_{\mathrm{net}}(i)$ is the net surface inflow to the stream in the $i$-th section, and $S_{\mathrm{in}}(i)$ and $S_{\mathrm{out}}(i)$ are the surface water inflow to and outflow from the stream in the section, respectively. In each section, we estimated groundwater discharge to the stream





based on the observed water balance with Eq. (2):

$$Q_{\mathrm{g}}(i) = Q(i) - Q(i-1) - S_{\mathrm{net}}(i) \tag{2}$$

where $Q_{\mathrm{g}}$ is the amount of discharge from the shallow aquifer to the stream, and $Q(i-1)$ and $Q(i)$ are the measured streamflows at the start and end points of section $i$, respectively.

### 2.2.2 Endmember mixing analysis using $^{87}$Sr/$^{86}$Sr and $\delta^{18}O$

Sr isotopes have been successfully used to identify the flow paths and sources of water (Shand et al., 2009). Strontium has four natural stable isotopes: $^{84}$Sr, $^{86}$Sr, $^{87}$Sr, and $^{88}$Sr, and the ratio of $^{87}$Sr to $^{86}$Sr (hereafter, $^{87}$Sr/$^{86}$Sr) has been widely used as a tracer to examine the sources and ages of rocks. The four Sr isotopes in rocks are variable because of the formation of radiogenic $^{87}$Sr by beta decay of naturally-occurring $^{87}$Rb, which has a half-life of $4.88 \times 10^{10}$ years (Faure and Mensing, 2009). Although minerals in rocks may have identical Sr isotope ratios at the time of formation, over time the decay of $^{87}$Rb to $^{87}$Sr leads to differences in $^{87}$Sr/$^{86}$Sr. In freshwater systems, the residence times of waters are short (days to $10^{2-3}$ years) compared to the half-life of $^{87}$Rb, and hence changes in $^{87}$Sr/$^{86}$Sr can be ignored. Unlike the light stable isotopes of water, the effect of kinetic fractionation on $^{87}$Sr/$^{86}$Sr caused by evaporation of water is negligible.

By combining the equation of mass conservation of water and Sr isotopes, the effect of mixing water samples with different Sr isotopic compositions and concentrations can be calculated with Eq. (3):

$$R_m = \frac{1}{C_m} \frac{C_a C_b (R_b - R_a)}{C_a - C_b} + \frac{C_a R_a - C_b R_b}{C_a - C_b} \tag{3}$$

where $R_m$, $R_a$ and $R_b$ are the $^{87}$Sr/$^{86}$Sr in the mixture, in A, and in B, respectively, $C_m$, $C_a$ and $C_b$ are the concentrations of Sr in the mixture, in A, and in B, respectively. Equation (3) indicates that the $^{87}$Sr/$^{86}$Sr of a mixture of two water samples A and B is a linear function of the inverse of the Sr concentration in the mixture (Faure and Mensing, 2009).

The existing ratio of water sample A in the mixture, $f_a$, can be calculated with Eq. (4):

$$f_a = \frac{R_m - R_b}{R_a - R_b} \tag{4}$$

Here, we used $^{87}$Sr/$^{86}$Sr and $\delta^{18}O$ for the calculation of the mixing ratios.

### 2.2.3 Estimation of groundwater discharge using stable isotopes of Sr and water

The groundwater discharge can also be estimated from The differential of groundwater volumes estimated with the observed streamflow, $Q(i)$, and ratios of groundwater in the stream, $f_{\mathrm{gw}}(\mathrm{Sr})$, and $f_{\mathrm{gw}}(\mathrm{O})$.

$$Q_{\mathrm{g}}(i) = f_{\mathrm{gw}}(i)Q(i) - f_{\mathrm{gw}}(i-1)Q(i-1) - f_{\mathrm{gw}}(i-1)\int S_{\mathrm{out}}(i) \tag{5}$$

We accounted only for cases where $S_{\mathrm{out}}$ is positive because the surface water inflow was presumably composed of surface irrigation water alone.





## 3 Results

### 3.1 Point-scale survey: changes in isotopic compositions through vertical percolation

#### 3.1.1 Sr isotopes

Figure 4 depicts the Sr isotopic composition of water sampled from a single rice paddy and its surrounding areas in August

2016. The numbers of samples from each location are shown in parentheses in the following summary: irrigation channel (1); drainage channels with concrete bottoms (2); drainage channel with soil bottom (1); outlet of rice paddy (2); inlet of rice paddy (1); spring (SAK) located next to the rice paddy (3); soil water at depths of 0.5 m, 1.0 m, and 1.5 m below the ground surface (1 from each depth). The irrigation rate was different for the two samples at the paddy outlet, i.e., $1.1 \ \mathrm{Ls^{-1}}$ for one sample and $0.06 \ \mathrm{Ls^{-1}}$ for the other (hereafter referred to as the higher and lower rates, respectively).

Water samples collected at the inlet of the rice paddy (light Green square in Fig. 4) had Sr isotopic compositions similar to irrigation water (yellow double circle), whereas water near the outlet (black squares) was characterized by higher $^{87}\mathrm{Sr}/^{86}\mathrm{Sr}$ ratios and lower Sr concentrations. Deviations of the paddy outlet data from the inlet datum can be explained by dilution of irrigation water with precipitation and by the condensing effect of evaporation. Precipitation was characterized by a high $^{87}\mathrm{Sr}/^{86}\mathrm{Sr}$ (0.70887) and low Sr concentration (1.54 $\mu\mathrm{gL^{-1}}$) and could not be plotted within the boundaries of Fig. 4. The

precipitation datum would lie on the extension of the dotted line in Fig. 4. The data for the two water samples collected from the outlet did not plot on the dotted line; this could be due to the condensing effect of evaporation. The dependence of Sr isotopic compositions of water at the paddy outlet on flow rate suggests that the isotopic compositions reflect the effects of evaporation to different degrees. The isotopic composition of water at the paddy outlet with higher rate plotted near the irrigation water because that water had less chance to mix with ponded water, whereas the isotopic composition of water at the

outlet with lower rate was further shifted from the original position because it had more opportunity to mix with ponded water.

During percolation of ponded water, $^{87}\mathrm{Sr}/^{86}\mathrm{Sr}$ ratios declined, and Sr concentrations increased with depth; $^{87}\mathrm{Sr}/^{86}\mathrm{Sr}$ and Sr concentrations were 0.70833 and 35.1 $\mu\mathrm{gL^{-1}}$ at 0.5 m, 0.70802 and 59.7 $\mu\mathrm{gL^{-1}}$ at 1.0 m, and 0.70749 and 66.1 $\mu\mathrm{gL^{-1}}$ at 1.5 m from the surface, respectively. $^{87}\mathrm{Sr}/^{86}\mathrm{Sr}$ at the shallowest depth (0.5 m) plotted close to the mean of the water at the paddy outlet (green circle), whereas the Sr composition of the deepest (1.5 m) soil water was similar to that of the spring water (red

asterisks). It should be noted that the water table at the location of soil water sampling was 1.67 m.

Water samples from the drainage channels with concrete bottoms (blue diamonds) had Sr isotopic compositions quite similar to those of the irrigation water. In contrast, water from the channel with a soil bottom (open diamond) had Sr composition that differed from that of irrigation water and plotted near the mixing line connecting irrigation water and spring (SAK) data. The fact that the isotopic composition of water from the drainage channel with a soil bottom plotted near the straight line on the

$^{87}\mathrm{Sr}/^{86}\mathrm{Sr}$-1/Sr diagram suggests that this water was a mixture of the end points of the straight line, i.e., irrigation water and shallow groundwater.

Overall, the Sr isotopic compositions collected from the point-scale survey suggest that water drained from the rice paddies with higher irrigation rates and that from the channel with a concrete bottom were unlikely to be changed because they had



less chance to mix with water with different Sr compositions. In contrast, the Sr isotopic compositions of water drained from rice paddies with lower irrigation rates and from channels with soil bottoms could change because that water had a chance to interact with water from different sources.

### 3.1.2   Water isotopes

Figure 5 shows the isotopic composition of water sampled during the point-scale survey. The isotopic composition of water at the inlet of the paddy was almost identical to that of irrigation channel water (yellow double circle), but the isotopic composition of water at the outlet (green square) differed substantially from that of water at the inlet, with the difference dependent on irrigation rate. The isotopic composition of water at the higher rate outlet was quite similar to that of the irrigation water and the paddy inlet, whereas water at the lower rate outlet was depleted in light isotopes, likely because of mixing with fractionated

water in the paddy. The difference in the isotopic composition of these two water samples was consistent with the observed difference in Sr isotope composition; that is, the higher the irrigation rate, the less chance the drained water had to mix with ponding water in the paddy, and vice versa.

The stable isotopic compositions of water in samples plotted linearly on a regression line of water sampled from wells in the watershed (the 'Well' solid line in Fig. 5), whereas the isotopic composition of water at the paddy outlet deviated slightly

from that line. The slope of the 'Well' line was 6.10 and deviated from the local meteoric water line (LMWL) at Utsunomiya, located approximately 20 km from the point-scale survey site (Yabusaki, 2010). The regression line for water in the paddy had a slope of 5.90 and deviated even more from the LMWL. These differences can be explained by differences in the influence of evaporation on shallow groundwater and rice paddies. Water in the rice paddy was much more extensively affected by evaporation than water in the soil and spring.

The isotopic compositions of soil water at depths of 1.0 m and 1.5 m were similar and plotted near the mean of the water at the paddy outlet. Unfortunately, the volume of the water sample from a depth of 0.5 m was insufficient for isotopic composition analysis. The isotopic compositions at shallow depths are generally subject to temporal variation, reflecting variations in isotopic compositions of recharged water; however, the invariance of water isotopic composition with lower depths suggests that the isotopic composition of water at a depth of 0.5 m may have been similar to that of water at depths of 1.0 m and 1.5 m.

This indicates that the isotopic compositions of water do not change as rapidly as Sr isotopic composition during vertical percolation from the rice paddy. The isotopic compositions of water in the drainage channels plotted along the 'Well' line to different degrees. The isotopic compositions of samples from channels with concrete bottoms were closer to the isotopic composition of irrigation water and water at the paddy inlet than the isotopic composition of the sample from the channel with a soil bottom, which plotted near the spring (SAK).





### 3.2 Watershed-scale surveys during irrigation and non-irrigation periods

#### 3.2.1 Sr isotopes

Figure 6 shows the Sr isotopic composition of samples collected from the watershed-scale survey during the irrigation period. The linearity of the plot of the isotopic composition of stream water (light blue square) on the $^{87}$Sr/$^{86}$Sr vs 1/Sr diagram from

upstream(Br-1) to downstream ends (Br-23) suggests that streamflow consisted of two endmembers. One potential endmember was water in the irrigation channel (yellow double circle) at the upstream end. Irrigation water had a higher $^{87}$Sr/$^{86}$Sr ratio and lower Sr concentration than those of the Kinu River water, from which irrigation water was diverted. The irrigation channel ran through rice paddies before reaching the Gogyo River watershed (Fig. 1). Water in the rice paddy, available only during the irrigation period, had a similar $^{87}$Sr/$^{86}$Sr to the irrigation water, but the Sr concentration was lower (Fig. 4), probably because

of dilution by precipitation. Changes in Sr composition from Kinu River water to irrigation water suggest that the irrigation channel received water drained from the surrounding rice paddies.

There do not seem to be any clear potential endmembers along the extension of the linear trend of the Gogyo River data in Fig. 6 because groundwater Sr compositions were scattered. However, the spring near the downstream end (spring (ODK), pink double square) could be the other endmember. The clustering of the isotopic compositions of wells located within 200 m of the

Gogyo River (red circles) and of the spring (ODK and SAK) around the isotopic compositions of water near the downstream end suggest that the Sr compositions of groundwater discharge were similar for these wells.

Figure 7 compares the Sr isotopic compositions of stream water during irrigation (light blue square) and non-irrigation (white square) periods. Streamflows at the upstream end were 0.985 $\mathrm{m^3s^{-1}}$ and 0.130 $\mathrm{m^3s^{-1}}$ during irrigation and non-irrigation periods, respectively. During both periods, the relationships between streamwater $^{87}$Sr/$^{86}$Sr and 1/Sr were linear. However,

Sr concentrations were higher and $^{87}$Sr/$^{86}$Sr ratios lower during the non-irrigation period. This pattern can be attributed to a decrease during the latter of the supply to the watershed of irrigation water, which has a relatively low Sr concentration and relatively high $^{87}$Sr/$^{86}$Sr.

#### 3.2.2 Water isotopes

Figure 8 shows the water isotopes, ($\delta^{18}$O and $\delta^2$H), from the watershed-scale survey during the irrigation period. The water

isotopes in the stream (light blue squares) were linearly related to each other from upstream (Br-1: lower-left) to downstream ends (Br-23: upper-right), and their abundance increased continuously in the downstream direction. The LMWL at Utsunomiya (dotted line, LMWL; Yabusaki, 2010), approximately 20 km from the watershed, had a slope of 8.30. The position of the precipitation datum below the LMWL was consistent with the observation that the water isotopic composition of precipitation tends to lie below the LMWL during relatively warm periods (Yabusaki, 2010; Yoshimura and Ichiyanagi, 2009; Tsuchihara et al.,

2016). The regression lines for stream water (short dashed line: 'Stream') and wells (long dashed line: 'Wells') were not as steep as the LMWL. Stream water and groundwater, enriched in heavy water isotopes, consisted of water affected by evaporation. The slopes of the 'Stream' and 'Wells' regression lines were 6.10 and 6.58, respectively. The difference between the stream relationship and the rice paddy datum (solid diamond) suggests that water in the rice paddies was most strongly fraction-





ated by evaporation and that stream water consisted of precipitation, irrigation water, and water drained from rice paddies. The water isotopes sampled at the wells (open circles) exhibited isotopic compositions similar to stream water, but their isotopic compositions were more depleted in heavy isotopes than those of the stream. It should be noted that samples from wells located within 200 m of the stream (red circle) were not clustered around spring water (ODK), suggesting that the latter (ODK) did

not have well-defined $\delta^{18}$O of discharged groundwater. The larger deviations of downstream water samples from the LMWL suggest that the effect of evaporation was higher in downstream than in upstream samples. The datum for the spring near the downstream end (ODK) was near the Br-23 datum. The irrigation water datum was near the Br-1 (upstream end) datum. These two samples could be endmembers of the stream, consistent with the conceptual model suggested by Sr isotope data.

Figure 9 compares water isotope compositions of stream water during irrigation and non-irrigation periods. Stream water

data were more scattered during the former (light blue square) than the latter (white square), but data from the non-irrigation period plotted along the regression line fit data from the irrigation period. Figure 9 also shows data for three water samples from the rice paddy (two samples from the point-scale survey in August, and one sample from the watershed-scale survey in June) (dashed line: paddy). The slope of regression line fit to the data from the paddy samples was 5.10.

### 3.2.3 Streamflow measurements

Figures 10 and 11 summarize the water balance in the six sections between the seven bridges (Br-1, 5, 9, 12, 15, 19, and 23). Table 1 summarizes streamflow measurement and estimated discharge from the aquifer to the stream. We measured the rates of lateral inflow at 30 channels (drainage from surrounding rice paddies) and of outflows at 22 channels (diversions from stream to paddies), and calculated the net surface inflow to the stream (black circles). We estimated groundwater discharge $Q_g$ using Eq. (2) for the six sections (red circles) from the residuals between increased/decreased volume of streamflow (blue circles)

and the net surface inflow $S_{net}$ in each section. We also calculated uncertainties of observed streamflow and inflow, assuming 15% observational errors (Carter and Anderson, 1963), and show the uncertainty bounds with shaded areas.

During the irrigation period, flow rates in the stream fluctuated because of water diversions and surface inflows to the stream from surrounding rice paddies (Fig. 10). The estimated discharges from the aquifer (red circles) also fluctuated, but not as much as streamflow and net surface inflow. The mean estimated discharge was 0.431 $m^3s^{-1}$. Discharge from the aquifer was

positive from Br-1 to Br-5 and from Br-12 to Br-19, but negative from Br-5 to Br-9 and from Br-19 to Br-23.

During the non-irrigation period, the flow rate increased continuously in the downstream direction, largely because of less influence from water diversions and drainage from rice paddies (Fig. 11). The volumes of net surface inflow were remarkably small compared to those during the irrigation period. The uncertainties of estimated groundwater discharges were therefore greater than during the irrigation period. There were net surface inflows only in sections Br-5 to Br-9 and Br-15 to Br-19. The

former inflow was a perennial stream with its headwaters at spring 'SAK' in the upper reaches (shown in Fig. 3). The latter inflow dried out in winter (from December 2016 through March 2017).



## 4 Discussion

### 4.1 Changes in isotopic composition of irrigation water via surface and subsurface drainage pathways of rice paddies

The point-scale survey revealed how these stable isotopes changed in two typical flow paths from rice paddies, namely surface drainage from rice paddies and subsurface drainage from recharged groundwater. Results indicate that changes in the $^{87}$Sr/$^{86}$Sr
of irrigation water were caused mainly by soil–water interactions below the surface. Changes in $^{87}$Sr/$^{86}$Sr via surface flow were not significant (Fig. 4). The values of $^{87}$Sr/$^{86}$Sr in soil water suggest that the rate of soil–water exchange was fast enough for the $^{87}$Sr/$^{86}$Sr of infiltrated water to become similar to the $^{87}$Sr/$^{86}$Sr of surrounding groundwater before it reached the water table. The values of $^{87}$Sr/$^{86}$Sr in groundwater were relatively stable in terms of space and time. Figure 7 indicated $^{87}$Sr/$^{86}$Sr values in the spring water (SAK and ODK) were nearly constant (0.7074) throughout the observation period, even though both
of the spring were located in the upstream and downstream of the watershed, respectively.

    The interpretation of changes in $\delta^{18}$O was more ambiguous. Water isotopes in rice paddy water changed due to kinetic fractionation associated with evaporation from the water surface, and the measured $\delta^{18}$O at the outlet suggests that the effects of kinetic fractionation resulted in different $\delta^{18}$O values that depended on the rate of irrigation (Fig. 5). Water at the higher irrigation rate paddy outlet had less chance to mix with ponded water, which was more affected by kinetic fractionation, and
vice versa. The isotopic composition of ponded water in the rice paddy was also affected seasonally by the height and spatial extent of rice plants because plant morphology controls evaporation from the water surface (Tsuchihara et al., 2017). While the stable isotopes in ponded water changed in many ways, percolation appeared to have little effect on water isotopes in subsurface flow, in contrast to the effects of percolation on the $^{87}$Sr/$^{86}$Sr of subsurface water. The $\delta^{18}$O values of soil water at different depths under the rice paddies were similar to the mean value of ponded water. However, the differences in the degree
of kinetic fractionation of soil water and ponded water (Fig. 5) suggest that percolated water consisted of ponded water that had been strongly affected by kinetic fractionation and water that had been less influenced by kinetic fractionation, probably irrigation water and precipitation. Although the above-mentioned processes changed the spatial and temporal distribution of water isotopes, in the spring (ODK) these were relatively constant (Fig. 9). This lower variability suggests that spring water isotopes were spatial and temporal averages.

Overall, the point-scale survey revealed that $^{87}$Sr/$^{86}$Sr values were distinctly different in surface and subsurface flow paths and could be used to represent both flow paths due to the relatively fast exchanges of Sr isotopes between irrigation and soil water. The $^{87}$Sr/$^{86}$Sr values in the spring near the Gogyo River channel were also almost identical throughout the observation period regardless of the location. The survey also revealed that $\delta^{18}$O values could be used as endmembers for surface and subsurface flow paths, albeit with some spatial and temporal variation associated with kinetic fractionation. Therefore, we
would argue that the Sr isotopes are more robust to quantify the groundwater contribution to the stream than the water isotopes.



## 4.2 Consistency of groundwater discharges estimated using different methods

### 4.2.1 Estimation of groundwater ratios in streamflow from endmember analyses

The watershed-scale survey indicated that streamflow of the Gogyo River consisted of two sources of water: irrigation water and discharged groundwater. We used $^{87}Sr/^{86}Sr$ and $\delta^{18}O$ for endmember analysis to quantify the relative contribution of
groundwater discharge to the stream.

The values of $^{87}Sr/^{86}Sr$ and $\delta^{18}O$ sampled from the wells varied depending on location and season. The spatial variation of $^{87}Sr/^{86}Sr$ and $\delta^{18}O$ were in the range of 0.020 (from 0.7065 to 0.7085) and 4.0 ‰ (from -10.4 ‰ to -6.4 ‰), respectively (Fig. 12). The blue shaded areas in Fig. 12 represent the approximate ranges of the temporal changes in the isotopic compositions between the irrigation and non-irrigation periods, 0.5 ‰ for the water isotopes and 0.00025 for $^{87}Sr/^{86}Sr$, which correspond to
12.5% of the spatial variations. The ratios of the samples plotted within the blue shaded areas in Fig. 12 to the total samples were 85% (31 out of 38 samples) for $^{87}Sr/^{86}Sr$ and 66% (23 out of 35 samples) for $\delta^{18}O$, respectively. Also, the higher correlation coefficient of $^{87}Sr/^{86}Sr$, 0.807, compared with that of $\delta^{18}O$, 0.551, indicated that the seasonal variation of the former was smaller than the latter. The point-scale survey revealed that $\delta^{18}O$ values of the paddy water that recharges the shallow aquifer may vary with the location and irrigation rate of paddies and that the isotopic compositions of water do not
change through vertical percolation. Thus, the spatial and temporal variations in $\delta^{18}O$ of the groundwater can be attributable to the difference in the effect of kinetic fractionation in the recharged water from rice paddies.

The groundwater that interacts with the stream is presumably water near the stream. The seasonal variations of $^{87}Sr/^{86}Sr$ in the water sampled from the wells within 200 m of the Gogyo River (red plots in Fig. 12) were small and clustered near the groundwater endmember (open circle), whereas those of $\delta^{18}O$ showed relatively large spatial and temporal variations. It is
worth noting that $^{87}Sr/^{86}Sr$ of the two springs in the upstream (SAK) and in the downstream (ODK) exhibited similar values (Fig. 7), whereas $\delta^{18}O$ of them were totally different (Fig. 9). Overall, the value of $^{87}Sr/^{86}Sr$ for the groundwater endmember was more stable spatially and temporally than those of $\delta^{18}O$. Therefore, we would argue that the groundwater ratios in the stream water estimated with Sr isotopes, $f_{gw}(Sr)$, are more robust than those estimated with water isotopes, $f_{gw}(O)$.

During the non-irrigation period, $f_{gw}(Sr)$ and $f_{gw}(O)$ increased across the river course, with their values and trends almost
identical (Fig. 13). Quite a good agreement was observed between $f_{gw}(Sr)$ ('Estimated with $^{87}Sr/^{86}Sr$') and $f_{gw}(O)$ ('Estimated with $\delta^{18}O$') during the non-irrigation period is encouraging because it is consistent with our conceptual model indicating that streamflow consists of irrigation water and groundwater discharge and suggests that the values assigned to endmembers were appropriate. The groundwater ratios at the upstream end were 38% ($^{87}Sr/^{86}Sr$) and 40% ($\delta^{18}O$) and were higher than those during the irrigation period. The same ratios at the downstream end, however, were lower (66% for $^{87}Sr/^{86}Sr$ and 69%
for $\delta^{18}O$) than during the irrigation period.

Figure 14 shows the ratios of groundwater to streamflow during the irrigation period. The values of $f_{gw}(Sr)$ ('Estimated with $^{87}Sr/^{86}Sr$') increased continuously in the downstream direction, from 0.25 at Br-1 to 0.79 at Br-23. Over the river course, the rate of increase was roughly constant, but we observed a slightly greater increase between Br-12 and Br-15 and a decrease between Br-21 and Br-22. The values of $f_{gw}(O)$ ('Estimated with $\delta^{18}O$') increased in a similar manner, but the values were in





most cases lower than $f_{\mathrm{gw}}(\mathrm{Sr})$ and ranged from 7% at Br-1 to 86% at Br-23. The rate of increase was relatively high between Br-8 and Br-9 and between Br-13 and Br-16. The latter increase was consistent with the pattern estimated with $^{87}\mathrm{Sr}/^{86}\mathrm{Sr}$.

The differences between $f_{\mathrm{gw}}(\mathrm{Sr})$ and $f_{\mathrm{gw}}(\mathrm{O})$ during the irrigation period can be attributed to uncertainty in the value assigned to the groundwater endmember during this period. The spring (ODK) was selected as the endmember for groundwater

based on the assumption that it represented the long-term mean of the stable isotopes of groundwater. However, the spatial and temporal variations in the values of $\delta^{18}\mathrm{O}$, sampled from the wells located within 200 m of the stream, have significant spatial variations and weak temporal variations, suggesting the groundwater endmember of $\delta^{18}\mathrm{O}$ has an uncertainty. Overall, , in watersheds with extensively irrigated rice paddies, we would argue that the use of Sr isotopes can be a robust tool to quantify groundwater discharge to streams compared to the use of water isotopes.

**4.2.2   Estimation of groundwater discharges to stream**

Figure 15 compares the groundwater discharge during the irrigation period estimated with three methods: the observed water balance in the stream sections, and the differential of groundwater volumes estimated from $^{87}\mathrm{Sr}/^{86}\mathrm{Sr}$ and $\delta^{18}\mathrm{O}$. The estimated groundwater discharge with two isotopes was more constrained than that with observed water balance and was within the uncertainty bounds of the observed water balance. The values estimated with $^{87}\mathrm{Sr}/^{86}\mathrm{Sr}$ and $\delta^{18}\mathrm{O}$ were in quite good agreement

especially in the lower part of the catchment. The inconsistency in the upper reach may reflect the uncertainties of endmembers of groundwater in $\delta^{18}\mathrm{O}$. While the estimated groundwater discharge with the water balance was negative in the sections between Br-5 and 9 and between Br-19 and 23, those estimated with two isotopes were positive throughout the river course. The estimated groundwater discharges were relatively higher in the lower reach (from Br-1 to Br-12), ranging from 0.052 to 0.193 $\mathrm{m}^3\mathrm{s}^{-1}$, compared to the upper reach (from Br-12 to Br-23), ranging from 0.183 to 0.345 $\mathrm{m}^3\mathrm{s}^{-1}$.

During the non-irrigation period, the estimated groundwater discharges with two isotopes were almost identical (Fig. 16), reflecting the consistency between $f_{\mathrm{gw}}(\mathrm{Sr})$ and $f_{\mathrm{gw}}(\mathrm{O})$ (Fig. 13). The estimated groundwater discharges were less than those during the irrigation period (note the differences of the ranges of vertical axes in Figs. 15 and 16) and almost no discharge occurred in the sections between Br-1 and Br-12, ranging from -0.052 to 0.038 $\mathrm{m}^3\mathrm{s}^{-1}$.

Interactions between groundwater and streams vary spatially and temporally (e.g., Keery et al., 2007). This study showed

that the Gogyo River is a typical gaining stream, but there was a clear difference in the volume of groundwater discharged from the aquifer during irrigation and non-irrigation periods. During the irrigation period, the groundwater discharge occurred throughout the river course, while it occurred only in the lower part of the watershed during the non-irrigation period. These differences in the volume and place of groundwater discharge may be a result of changing hydraulic gradients due to groundwater head variation, reflecting the decrease in irrigation water supply and precipitation in the non-irrigation period. The raised

water tables during the irrigation period enhanced the groundwater discharge throughout the watershed, including the upper part of catchment, and increased the volume of groundwater discharge.

From the perspective of water management, the temporal changes in groundwater discharge from the aquifer could play an important role, especially during drought periods, given that surface flow during droughts can be less compared to during wet periods because of higher evaporation losses and the inclination of farmers to restrict surface drainage (Yoshida et al., 2016).





However, observations of groundwater discharge from agricultural areas have mainly relied on streamflow measurement, and were carried out when the system was regarded as being in a steady state (Yu-Chuan et al., 2016; Fan et al., 2013), i.e., with constant irrigation supply, negligible effects of precipitation, and outflow from the irrigation system assumed to be composed of only the return flow of irrigation water. The estimated groundwater discharge is thus limited to constant values when the system is in steady state. We would argue that the use of $^{87}$Sr/$^{86}$Sr can be a novel and robust tool to quantify temporal changes in groundwater discharge especially in watersheds substantially disturbed by rice paddy irrigation.

## 5    Conclusions

This study used three different methods (streamflow observation, stable isotopes of water, and Sr isotopes) to quantify groundwater discharge to streams in a watershed substantially impacted by rice paddy irrigation in central Japan. We conducted point- and watershed-scale observations of surface water, soil water, groundwater, and ponded water in rice paddies and examined how isotopic compositions of strontium and water changed through water flux in the watershed.

The point-scale survey revealed how these stable isotopes changed in two typical flow paths from rice paddies, namely surface drainage from rice paddies and subsurface drainage from recharged groundwater. The point-scale observations revealed that Sr isotopes were more appropriate for quantification of groundwater discharge, because the groundwater end member was well defined and significantly different from surface water due to relatively fast soil–water interactions below the surface. The interpretation of changes in $\delta^{18}$O was more ambiguous because ponded water in rice paddies were strongly affected by kinetic fractionation, irrigation rate, and plant morphology.

The watershed-scale observations showed that both stable isotopes were of the stream water changed linearly from upstream end to downstream end, suggesting streamflow consisted of two endmembers. We then quantified groundwater discharge to the stream based on three methods: the water balance in the stream sections, and the differential of groundwater volumes estimated from the stable isotopes of Sr and water. The water balance method, based on The measurement of streamflow and surface lateral inflow/outflow, yielded large uncertainty in the estimation due to the observation errors in streamflow measurement, while the stable isotopes provided well constrained estimates during both irrigation and non-irrigation periods.

The ratios of groundwater in the stream, estimated from Sr isotopes, fell in the range of 7–86% during the irrigation period and 38–66% in the non-irrigation period. Stable isotopes of water also provided good estimates during the non-irrigation period, but underestimated groundwater discharge during the irrigation period due to the ill-defined groundwater end member. Sr isotopes, therefore, can be a robust way for quantifying the groundwater discharge, especially in watersheds where the direct use of water isotopes was hindered by their evaporative enrichment from the water surface. Because of temporally constant Sr isotopes values in groundwater, temporal variations in interactions of stream and groundwater can be addressed using Sr isotopes as a tracer.

*Competing interests.*   The authors declare that they have no conflict of interest.



*Acknowledgements.* This research is supported by JSPS KAKENHI Grant Number JP 16K18774. This study was conducted with the support of the Joint Research Grant for the Environmental Isotope Study of Research Institute for Humanity and Nature.





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





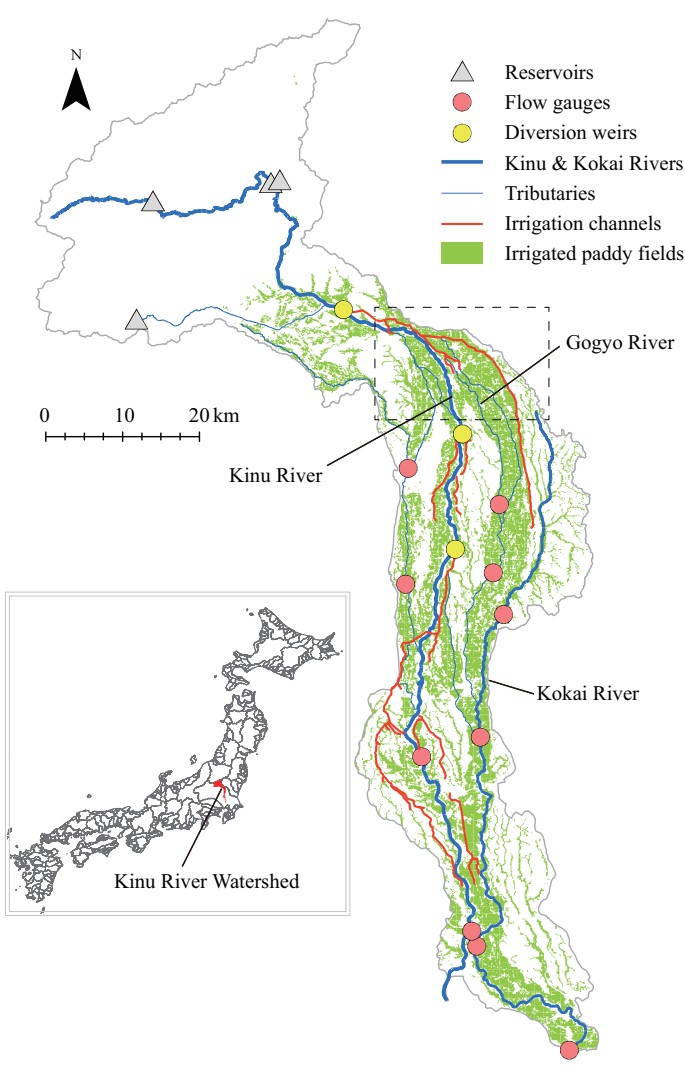

**Figure 1.** Overview of Kinu River watershed. The outlined region is depicted in Fig. 2.

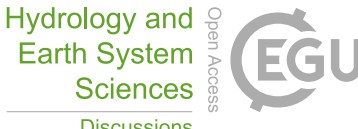



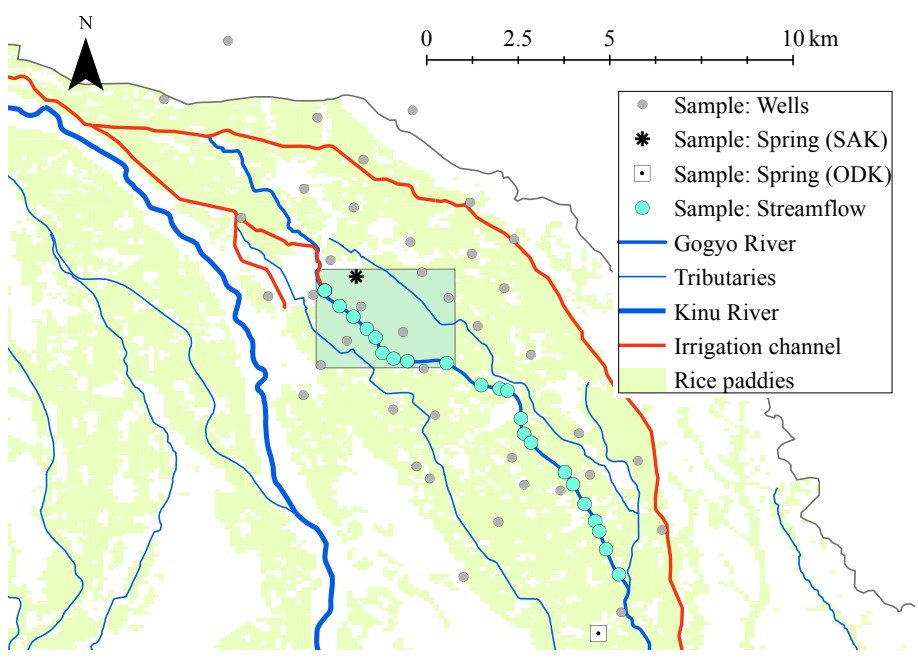

**Figure 2.** Overview of the Gogyo River watershed and sampling locations. The outlined region is shown in Fig. 3.



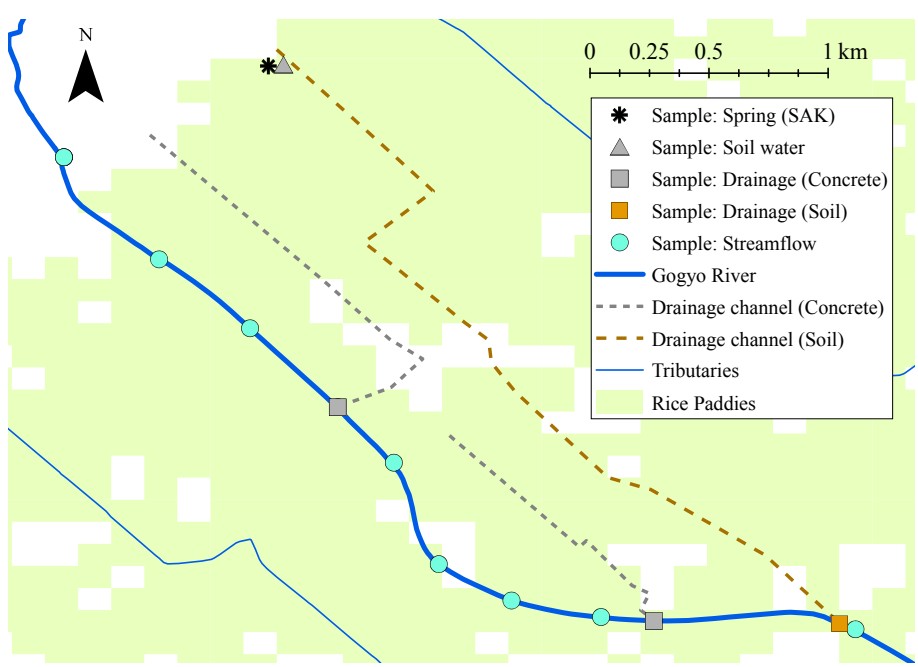

**Figure 3.** Sampling locations for the point-scale survey. Location is shown in Fig. 2.





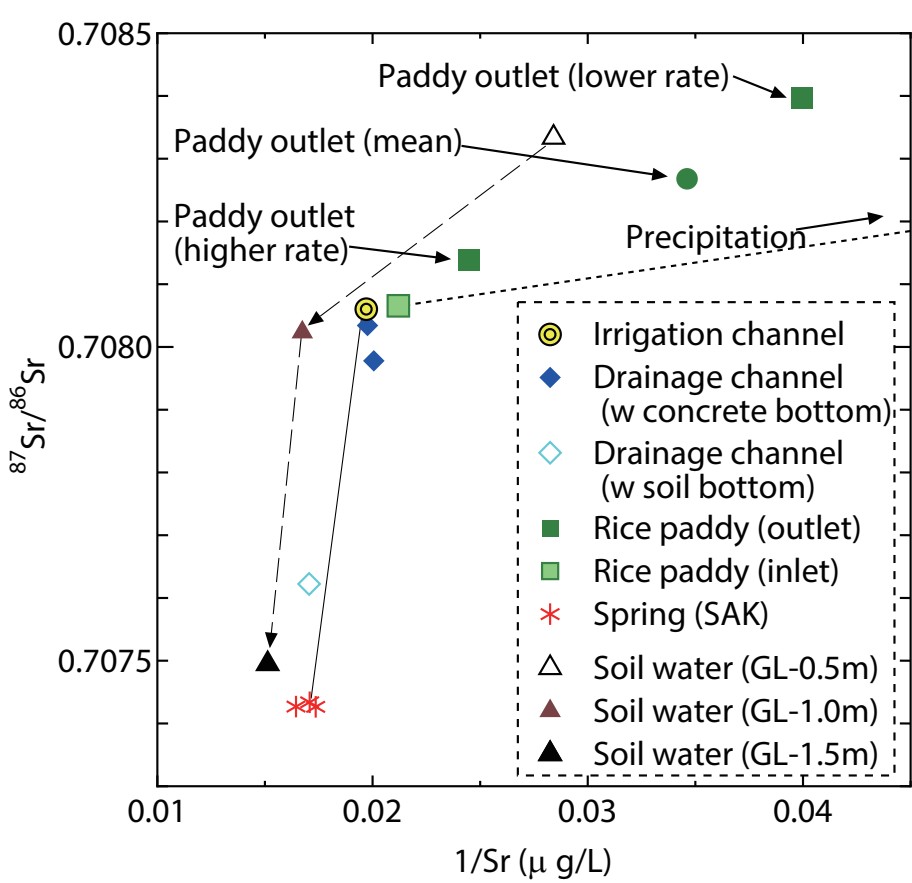

**Figure 4.** $^{87}$Sr/$^{86}$Sr–1/Sr diagram for the point-scale survey.



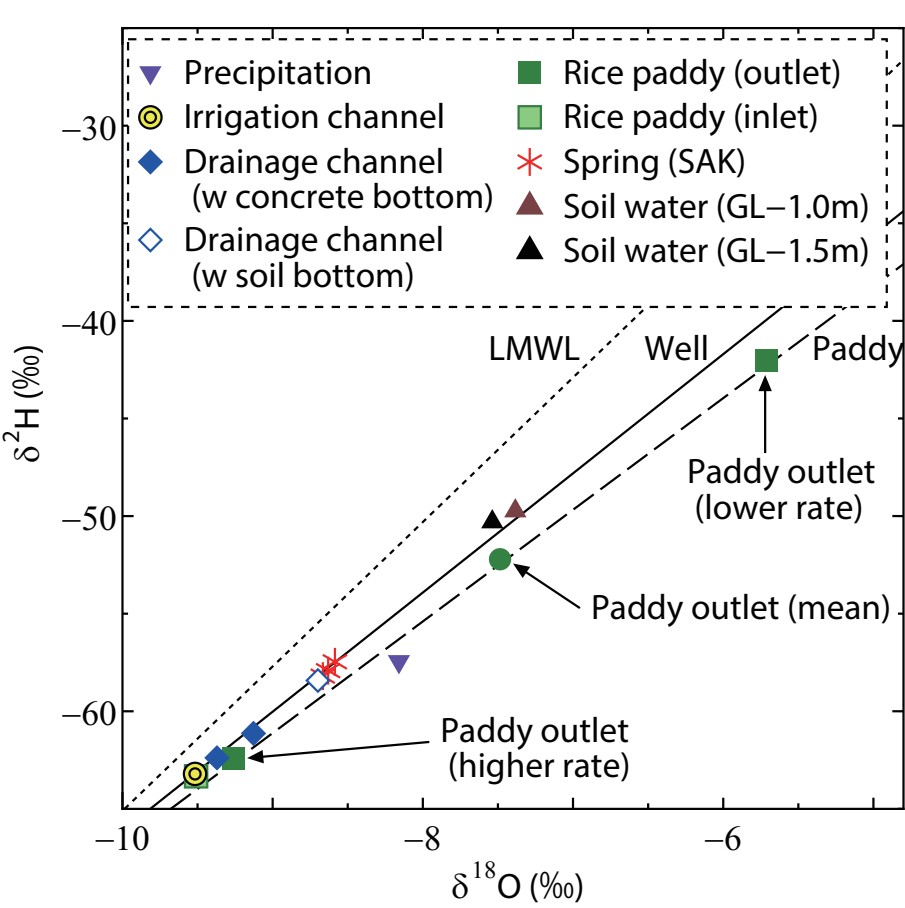

**Figure 5.** Water isotope diagram for the point-scale survey.





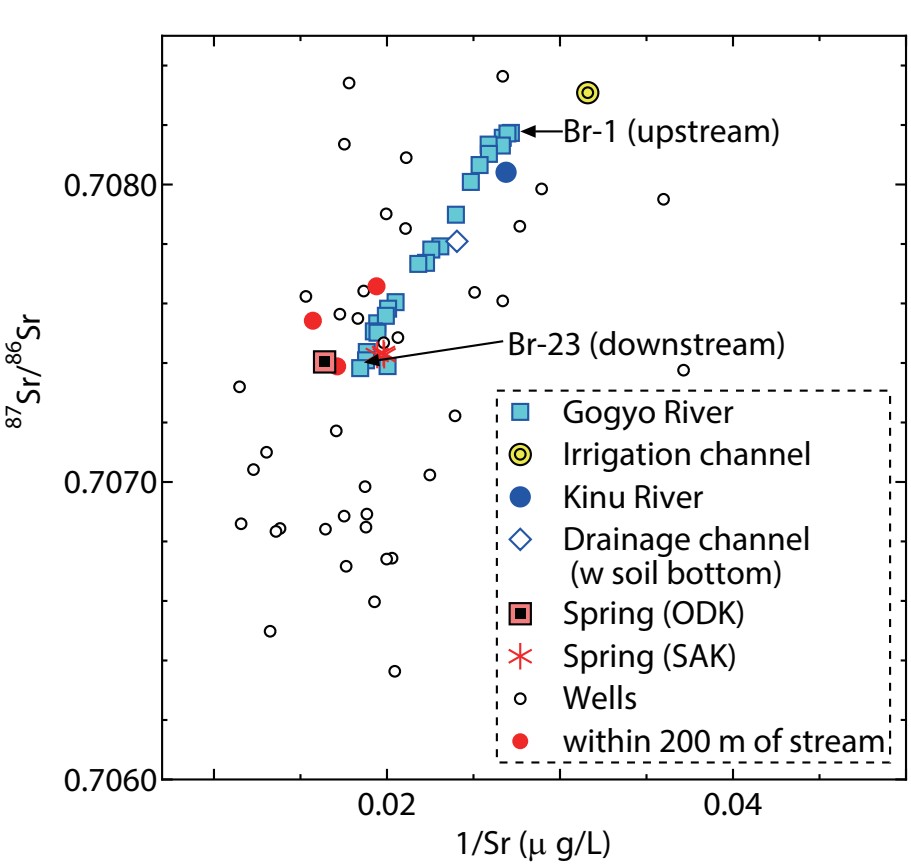

**Figure 6.** $^{87}$Sr/$^{86}$Sr–1/Sr diagram for the watershed-scale survey during the irrigation period.





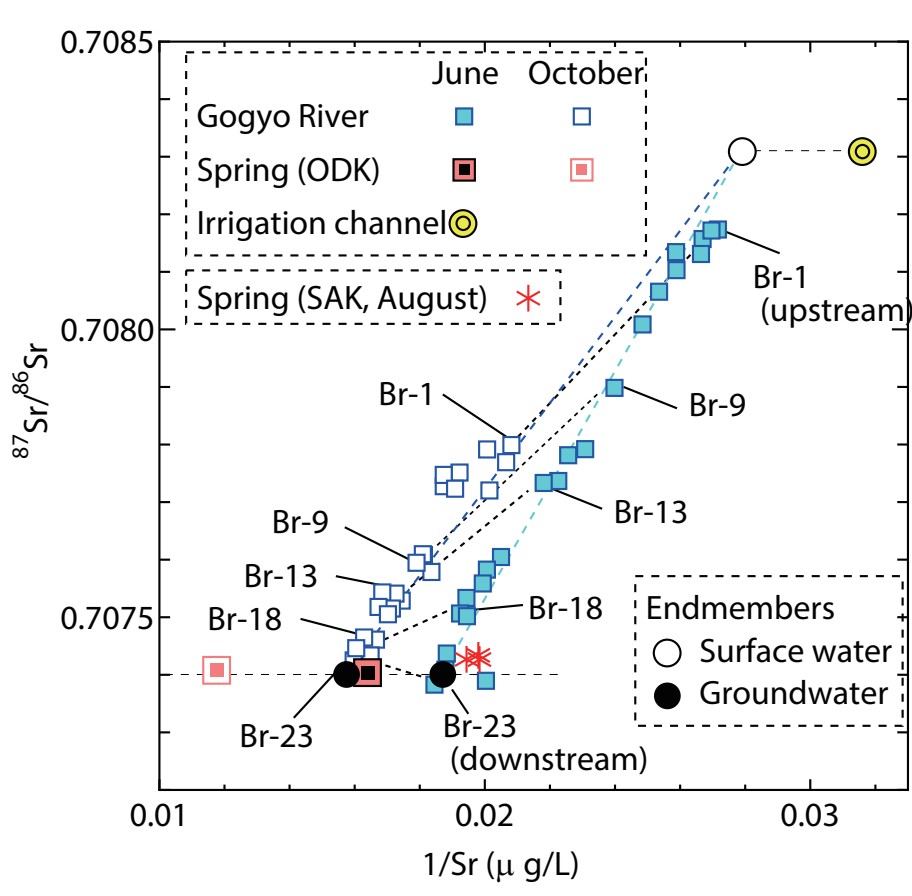

**Figure 7.** Comparison of streamflow $^{87}$Sr/$^{86}$Sr–1/Sr during irrigation and non-irrigation periods.





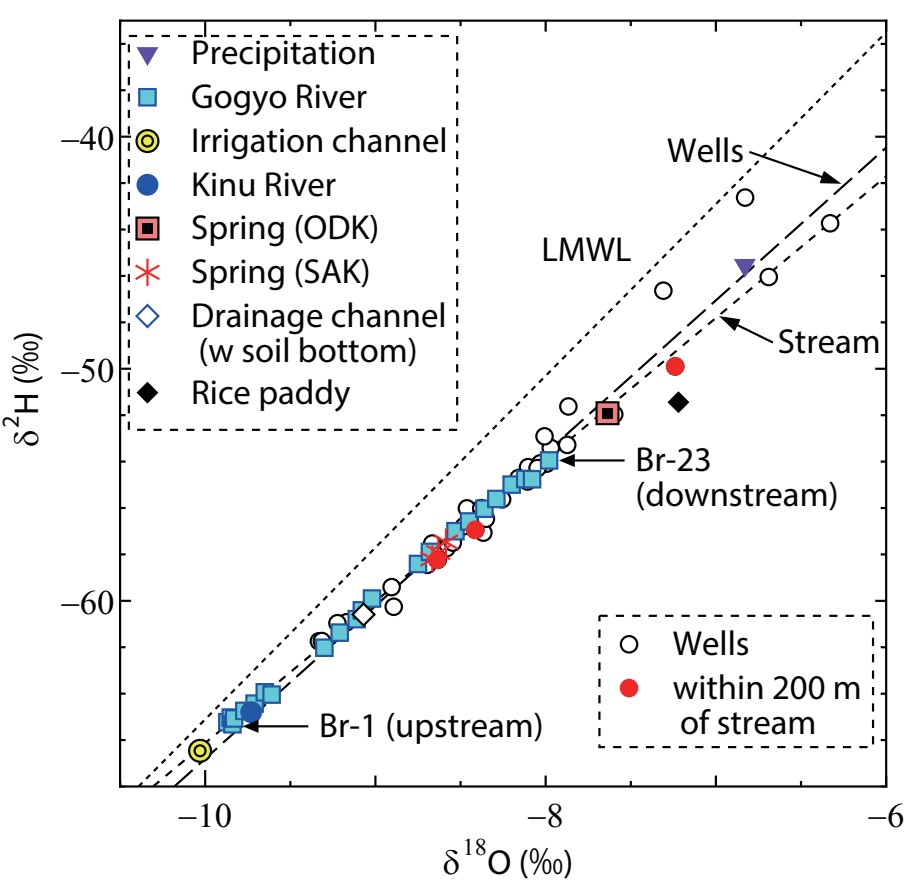

**Figure 8.** Water isotope diagram for the watershed-scale survey during the irrigation period.





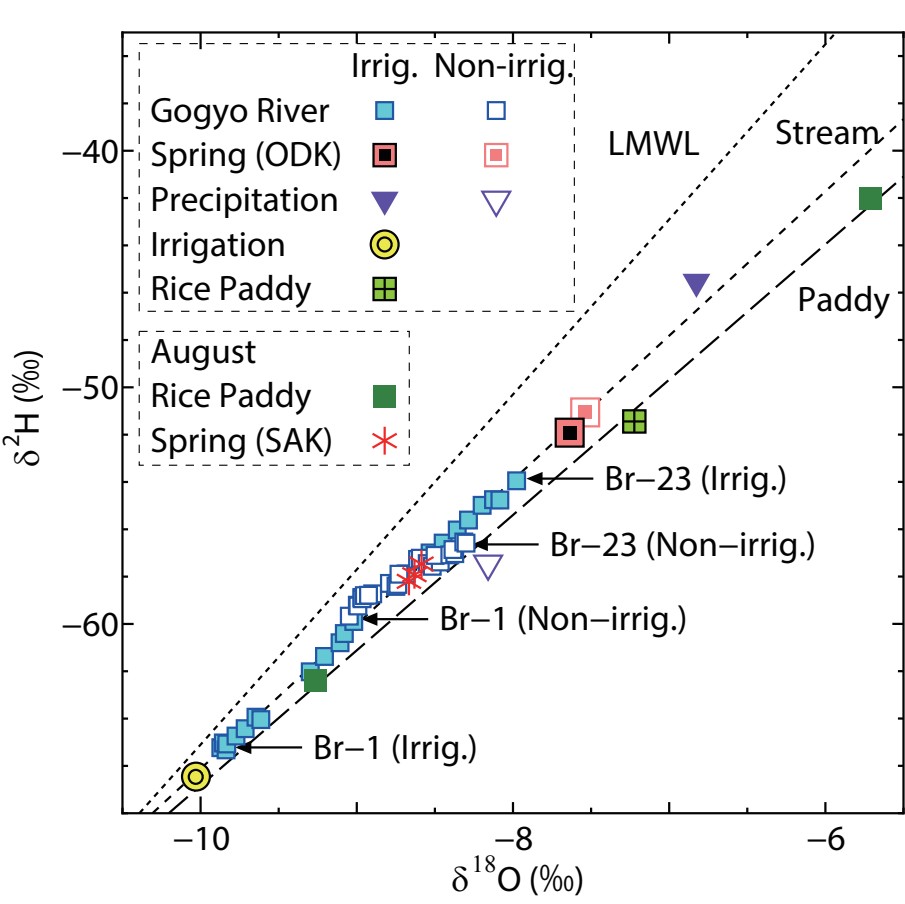

**Figure 9.** Comparison of water isotope diagrams for irrigation and non-irrigation periods.





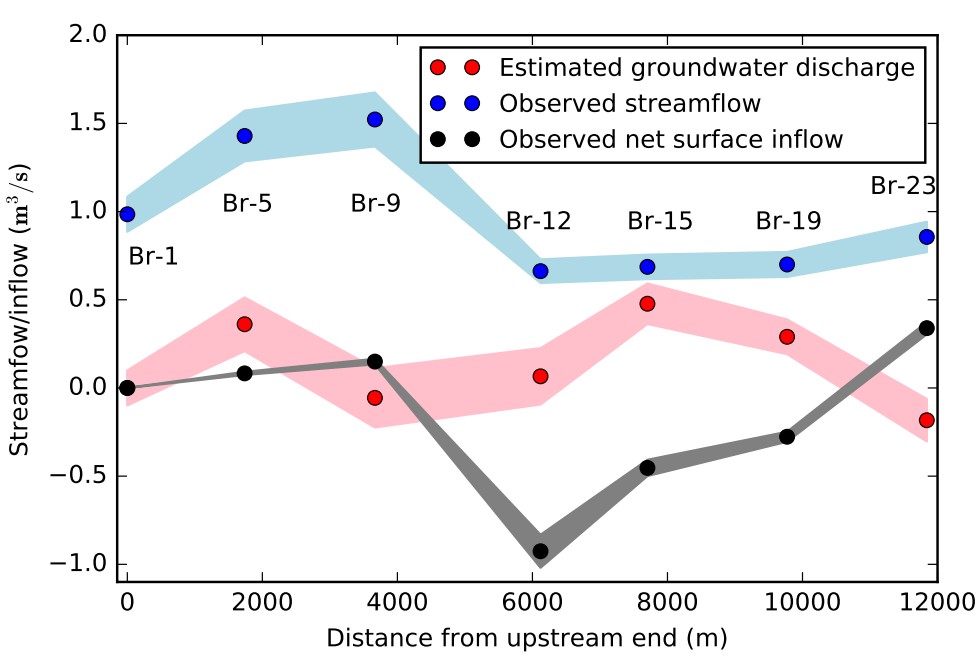

**Figure 10.** Estimated groundwater discharge to stream based on the observed water balance during irrigation period.





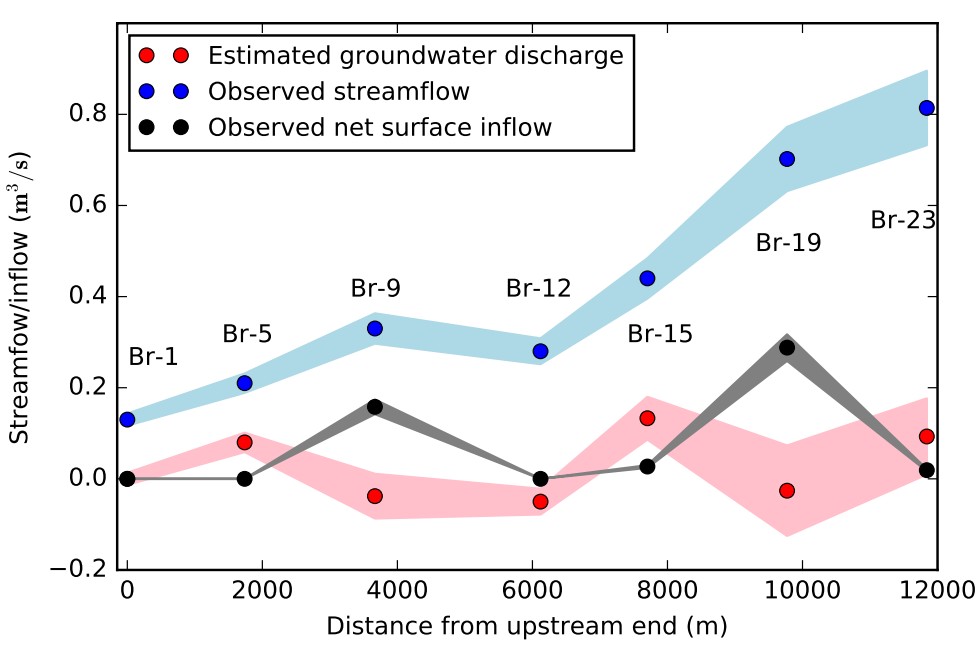

**Figure 11.** Estimated groundwater discharge to stream based on the observed water balance during non-irrigation period.





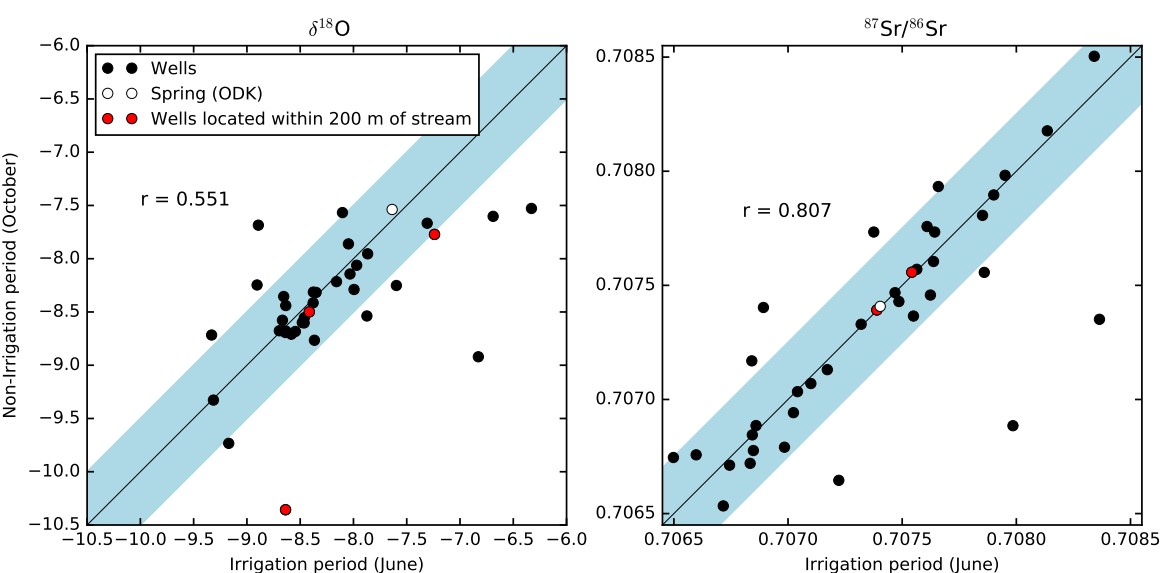

**Figure 12.** Values of $\delta^{18}$O and $^{87}$Sr/$^{86}$Sr in the groundwater (wells): comparison in the irrigation and non-irrigation periods.





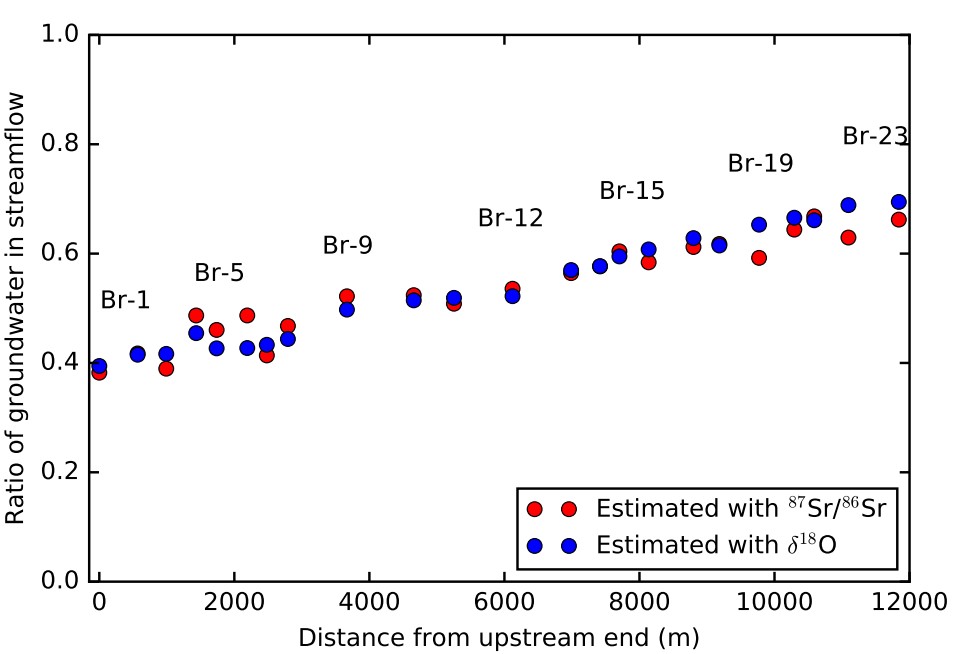

**Figure 13.** Ratios of groundwater in the streamflow based on $\delta^{18}O$ and $^{87}Sr/^{86}Sr$ during non-irrigation period.





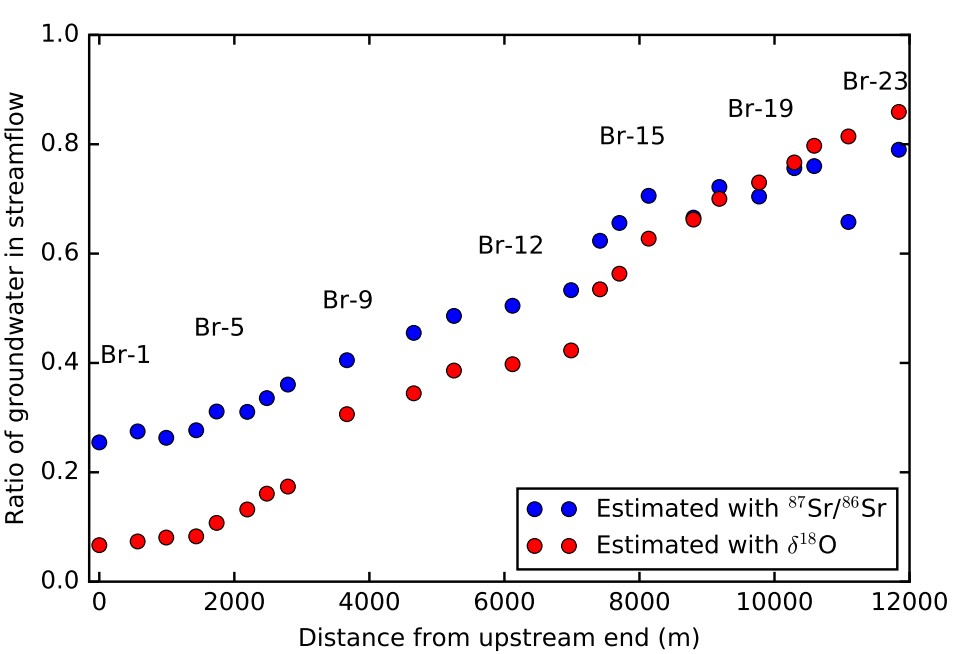

**Figure 14.** Ratios of groundwater in the streamflow based on $\delta^{18}$O and $^{87}$Sr/$^{86}$Sr during irrigation period.





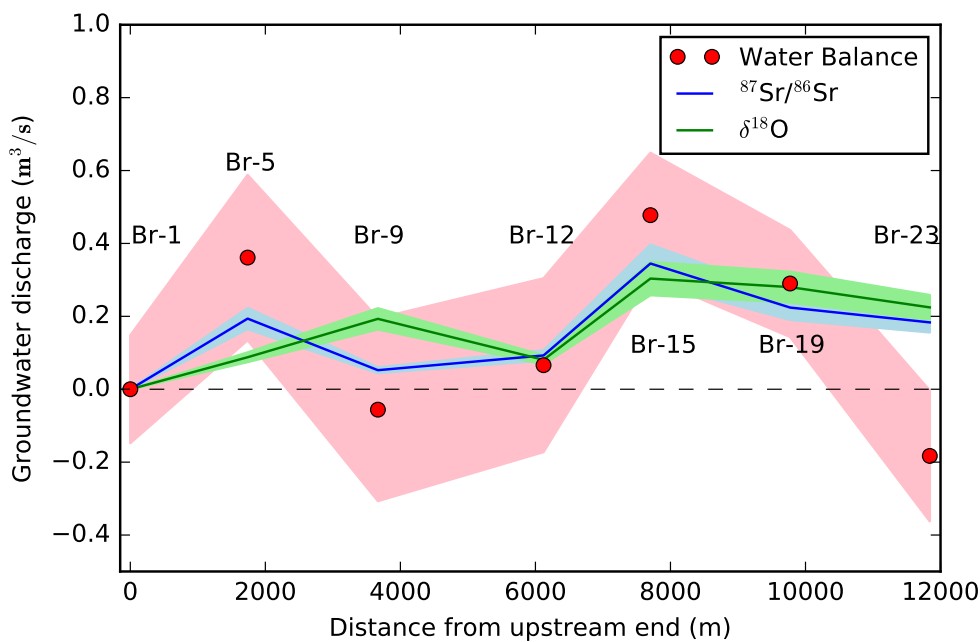

**Figure 15.** Comparison of the groundwater discharges based on three different methods: observed water balance, $\delta^{18}$O and $^{87}$Sr/$^{86}$Sr during irrigation period.





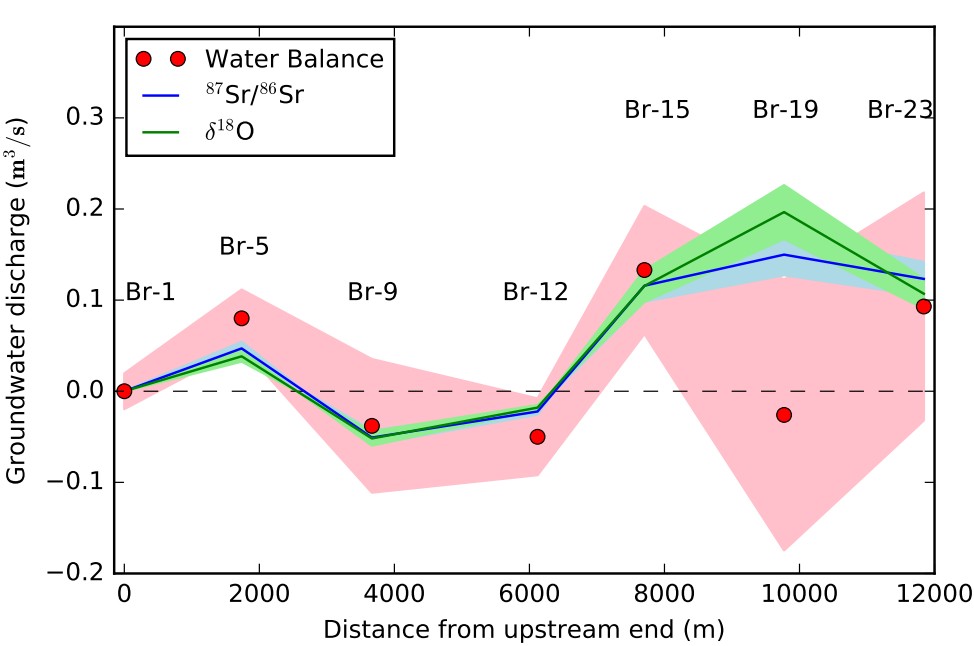

**Figure 16.** Comparison of the groundwater discharges based on three different methods: observed water balance, $\delta^{18}$O and $^{87}$Sr/$^{86}$Sr during non-irrigation period.





**Table 1.** Summary of streamflow observations and estimated groundwater discharge

| Name | Distance | Irrigation period (June, 2016) | | | Non-irrigation period (October, 2016) | | |
|------|------|------|------|------|------|------|------|
| | (m) | Streamflow $(m^3s^{-1})$ | Groundwater in-flow $(m^3s^{-1})$ | Net surface in-flow $(m^3s^{-1})$ | Streamflow $(m^3s^{-1})$ | Groundwater in-flow $(m^3s^{-1})$ | Net surface in-flow $(m^3s^{-1})$ |
| Br-1 | 0 | 0.985 | - | - | 0.130 | - | - |
| Br-5 | 1738 | 1.429 | 0.361 | 0.083 | 0.210 | 0.080 | 0.000 |
| Br-9 | 3667 | 1.522 | -0.056 | 0.150 | 0.330 | -0.038 | 0.158 |
| Br-12 | 6120 | 0.662 | 0.066 | -0.926 | 0.280 | -0.050 | 0.000 |
| Br-15 | 7703 | 0.687 | 0.478 | -0.453 | 0.440 | 0.133 | 0.027 |
| Br-19 | 9772 | 0.700 | 0.290 | -0.277 | 0.702 | -0.026 | 0.288 |
| Br-23 | 11841 | 0.856 | -0.183 | 0.339 | 0.814 | 0.093 | 0.019 |