# Peer review of "Quantification of seasonal variabilities in groundwater discharge in an extensive irrigation watershed using H, O, and Sr isotopes"

_Hydrology and Earth System Sciences, 2018_

## Referee Comment (RC1) · Anonymous Referee #1 · 2 Jan 2019

The authors present a study about quantification of groundwater discharge in a river in an extensive irrigation watershed in Japan. Their study used three methods to estimated the groundwater impact: flow measurements and 2 isotopic approaches (stable isotopes of the water molecules and St isotopes). The global approach and the sampling strategy is good and appropriate to answer the question (the raw data are not provide and should be added as supplementary material). This study is of interest for the scientific community and also as potential tools in terms of water management. The text is relatively short considering the work presented and the various approaches applied. Thus, the detail reasoning especially concerning the groundwater contribution estimated through the 3 approaches is hard to follow and thus the main conclusions are

too weakly supported by the main text. The manuscript would benefit to have a more detailed text, and to remove at least one or two figures (e.g. figures 6 and 12). The discussion section should be more detailed and argued with a solid comparison of both isotopic approaches considering their discriminating power. Sr is a WRI marker and thus will translate the lithological variations and water circulations, to be useful, contrasted signatures of the considered end-members are required; while stable isotopes of the water molecule, will mainly trace (in this context) the evaporation effect and could highlight variations along the hydrological cycle. For these reasons, and comments below, I recommend this work to be published in HESS with major revisions.

Specific comments :

Almost all the references are cited in the introduction. The main text, and especially the discussion, should refer to appropriate references. Note that only 23 references are cited, which is not enough considering the 3 applied approaches and the abundant available literature available for each approach.

Method section:

Samples dedicated to Sr isotopes analysis must be filtered and acidified to pH 2 with suprapure HNO3. Authors probably do it this way and this should be specified.

Sr isotopes analysis : change 87Sr/86Sr of 8.37. . .. to 88Sr/86Sr (page 4,line 17).

Sr concentration analysis method is not reported: please add it with the uncertainty.

Water isotopes analysis too poorly explained: especially using laser method, the salinity effect of the samples should carefully considered. Here we have no idea of the electric conductivity or TDS of the samples to evaluate a potential impact.

More generally, the manuscript would benefit to have a brief overview of the major elements concentration.

For water isotopes, the uncertainty for both O and H measurements should be added.

Section 2.2.1 : specify the measurements period, over a same day, week , . . . ? what as the weather during that period ?

Section 2.2.2 : Equation 3 is not very clear, why not simply using for [Sr] and Sr isotopes (with R = 87Sr/86Sr , C = Sr concentration, f = fraction of endmember 1) : Rm = [f(R1*C1)+ (1-f)(R2*C2)] / Cm and Cm = f C1 + (1-f) C2 Specify that equation 4 is only true for stable isotopes of the water molecule.

Section 3.1.2 : line 9 : . . . because of mixing with fractionated water . . . Could we also consider that water has undergone direct evaporation and not only a mixing with an "older" evaporated water ?

Section 3.2.1 : in this section, the choice of the groundwater end-member (ODK / SAK) need to be better explained and argued as we note that the local wells present great variations for Sr isotopic signatures (figure 6). In the same way in figure 7 and the text referring to this figure, the end-members "surface water" derived from the irrigation channel and the "groundwater end-member(s)" derived from the springs SAK and ODK should be explained.

Section 3.2.2 : lines 25-30 : data plotting below the LMWL have an explanation, the studies cited give this explanation, it's not only an observation. Line13 page 9 : data from the paddy define a line with a slope of 5.1, is this value in agreement with local annual humidity ?

Section 4.1 : line 17 : specify why percolation has no effect on sable isotopes of water molecule. Lines 23-24: not clear, do you mean that spring (ODK) water reflect the buffering effect of the aquifer ?

Section 4.2.1 : lines 6 to 16 : this part can be shortened (too descriptive in the discussion section) . Figure 12 can also be removed. Line 31 and followings : How is calculated the fraction of groundwater estimated from Sr isotopes in Br 1 and Br 23 ? from figure 7 , Br1 is constituted of less than 10% of groundwater and Br23 has the

signature of the groundwater end-member, i.e. almost 100% of groundwater . . . Please clarify. Same thing for the global calculations in figure 13 and 14. In figure 14, colors are inversed compared to figure 13, to be checked.

---

## Referee Comment (RC2) · Anonymous Referee #2 · 7 Jan 2019

General Comments ‒The manuscript attempts to quantify seasonal variabilities in groundwater discharge in an extensive irrigation watershed using H, O, and Sr isotopes. conducted point- and watershed-scale observations of surface water, soil water, groundwater, and ponded water in rice paddies and examined changes in these isotopic compositions. They conclude that the ratios of groundwater to the stream was in the range 7–86% during the irrigation period and 38– 66% during the non-irrigation period. The use of Sr isotopes showed better results that those of stable water isotopes.

Âă‒The manuscript contains some useful material, however in its current form the manuscript is not publishable. It seems particularly apparent given the amount of time

the manuscript spends describing the measurement results itself. While no quantitatively evidence to support their assumptions, e.g. the water isotope diagram can not provide direct evidence. And despite the length there are several statements in the technical description which need to be clarified as they indicate some further analysis is required to confirm the results.

Major comments:

1. I don't find the new insight from this paper. Since Sr isotopes is less fractionation, it is well known that the use of Sr isotopes has the higher potential to aid in quantification of temporal variations in groundwater discharge.

2. The manuscript is not straightforward, and the results are difficult to understand. For me, it is better to show the sampling locations in more detailed way. I can not find where is the location of Br1- 23. And where is the upstream and downstream mentioned in the manuscript. Also, please give the detail information about sampling date. This is extremely important for stable water isotope study. If surface water, soil water, groundwater, and ponded water in rice paddies are sampling in different days, the authors should make sure they are not change significantly in temporal scale.

3. Ponded water isotope in rice paddies indeed showed large spatial variability. The inflow side and outflow side will show large differences. I don't know whether the authors consider this or not. To get an average value, I think it need special treatment.

4. The most important thing is neglecting the effect of precipitation. Please plot out the precipitation during the sampling period. Precipitation will definitely change all the results.

5. The two endmembers partitioning method is good but neglecting the recharge process. This may be important for irrigation period. Large portions of irrigation water may recharge regional aquifer. Any idea or evidence?

6. To give some quantitative conclusions from the isotope experiment, I suggest the

authors should at least discuss the result with water balance components (precipitation, ET, irrigation, flow rates in rivers, groundwater table fluctuations, etc.) at the specific studied area. For instance, the authors can estimate the irrigation water based on the local irrigation schedule and the cultivated area. Currently, the authors only present the peak flow rates for the whole diversion weirs (71 mˆ3/s), it is hard for us to link this to your experimental results.

Minor comments: Line 24 p5: The – the

Line 25 p6: water table was 1.67 m — this is confusing. Do you mean groundwater depth?

Line 21 p9: Usually observation error is not portable, given the data features at different basins are quite different. Moreover, the reference you cite is from 1963, now we have more accurate and convenient method to measure the flow rate.

Lines 6, 8 in p11: leave a space after ‰

Line 7 p12: duplicated ,

Line 21 p13: The – the

Line 26 p13: please use endmember or end member consistently in the paper.
* * *

---

## Author Comment (AC1) · 19 Feb 2019

Reply to comments from Anonymous Referee #1.

**General comment:**

*The authors present a study about quantification of groundwater discharge in a river in an extensive irrigation watershed in Japan. Their study used three methods to estimated the groundwater impact: flow measurements and 2 isotopic approaches (stable isotopes of the water molecules and St isotopes). The global approach and the sampling strategy is good and appropriate to answer the question (the raw data are not provide and should be added as supplementary material). This study is of interest for the scientific community and also as potential tools in terms of water management. The text is relatively short considering the work presented and the various approaches applied. Thus, the detail reasoning especially concerning the groundwater contribution estimated through the 3 approaches is hard to follow and thus the main conclusions are too weakly supported by the main text. The manuscript would benefit to have a more detailed text, and to remove at least one or two figures (e.g. figures 6 and 12).*

**[Response]**

We appreciate the reviewer for his/her time and effort in reviewing our manuscript as well as the corresponding thoughtful comments.

As suggested by the reviewer, we will present raw data as supplementary material; however, the detailed analysis on the principal component was out of the scope of this study. We are currently working on multiple geochemical tracers to understand the hydrological processes in this watershed and it will be reported in our following paper.

In the previous version of the manuscript, the method of determining the endmembers was not clearly described, and there was an inconsistency between the endmember described in the manuscript (i.e., the values of the irrigation water and the spring, ODK) and depicted in Fig 7 of the previous manuscript (i.e., the plot of the endmember indicated by the open and black circles). In the previous manuscript, we calculated the ratio of groundwater to the stream water using the former (the values described in the manuscript), instead of the latter (the values depicted in the figure). This inconsistency induced confusion in the readers.

Therefore, we created a new section (3.3.1) describing the endmember determination and recalculated the ratio of groundwater in the stream (3.3.2) and the quantified groundwater discharge (3.3.3). More specifically, we determined the groundwater endmember of Sr isotope from the crossing point of the regression line of the stream water samples and the horizontal line

departing from the spring, ODK in Fig 4 (b) (in the revised manuscript). We describe the response in more detail in the following specific comment.

[Figure]

Figure 4: Sr isotope obtained from the watershed-scale survey: (a) all samples obtained during the irrigation period, and (b) comparison of Sr isotopic compositions in stream water during irrigation and non-irrigation periods.

Figure 12 (in the previous manuscript) is an important figure showing that the spatial and temporal variations in Sr isotopes was less than those of water isotopes. These figures support part of our conclusion that Sr isotopes are more appropriate than water isotopes in terms of the consistency for representing the groundwater endmember; thus, we would like to retain these figures in the revised manuscript.

*The discussion section should be more detailed and argued with a solid comparison of both isotopic approaches considering their discriminating power. Sr is a WRI marker and thus will translate the lithological variations and water circulations, to be useful, contrasted signatures of the considered end-members are required; while stable isotopes of the water molecule, will mainly trace (in this context) the evaporation effect and could highlight variations along the hydrological cycle. For these reasons, and comments below, I recommend this work to be published in HESS with major revisions.*

**[Response]**

We updated sections 4.1 and 4.2 to describe the testing of the robustness of the endmembers and the consistency of estimated groundwater ratios in the stream using two isotopes. We also added chloride concentration to corroborate our conclusion as described below.

By using the abovementioned endmembers consistently throughout the manuscript, the estimated values of groundwater ratios in the stream in the previous manuscript have been revised (see figures below). In the previous manuscript, the estimated ratios from Sr and water isotopes were in good agreement in the non-irrigation period, while they differed in the irrigation period (see updated version of the Figures 13 and 14). However, in the revised manuscript, they show good agreement during the irrigation period, whereas discrepancies were observed during the non-irrigation period. The groundwater ratios in the stream, estimated from Sr isotopes, increased over the river course: 6–93% during the irrigation period and 46–99% in the non-irrigation period. Stable isotopes of water provided similar estimates during the irrigation period (7–83%), but discrepancies were observed during the non-irrigation period (41–69%).

[Figure]

Figure 13 and 14 [Updated version] Ratios of groundwater in the streamflow based on $\delta^{18}$O and $^{87}$Sr/$^{86}$Sr during irrigation period (left) and non-irrigation period (right).

To elucidate the causes of the differences in the estimates from both isotopes, we used the chloride concentration of stream water. Chloride concentration at the upstream end was 5.24 mg/l (irrigation) and 5.09 mg/l (non-irrigation) and showed a similar increasing trend up to the middle reach of the stream. However, the concentration differed at the downstream end: 7.23 mg/l (irrigation) and 6.25 mg/l (non-irrigation). The chloride concentration of the spring ODK, located near the downstream end, was 7.47 mg/l and 7.62 mg/l during the irrigation and non-irrigation

periods, respectively.

The higher chloride concentration at the downstream water and its similarity to the spring ODK in the irrigation periods suggests that the stream water was highly affected by groundwater, which was consistent with the estimated groundwater ratios in the stream using either Sr or water isotopes. On the other hand, the lower chloride concentration at the downstream end in the non-irrigation period suggested that the stream water was less affected by groundwater discharge. This observation was consistent with the estimates of groundwater ratio using water isotopes, corroborating the overestimation by Sr isotopes.

[Figure]

Figure 6 [added to the revised manuscript] Chloride concentration in stream water and the spring, ODK.

Attention should be paid to distinguishing groundwater discharge and hyporheic exchange (Kalbus et al., 2006). In streams, the ratio of groundwater discharge estimated using Sr isotopes might not entirely be attributed to groundwater discharge, but include water-rock interaction (WRI) in hyporheic zones. During the non-irrigation period when the groundwater discharge was not dominant due to low groundwater level, the hyporheic exchange increases the probability of stream water interacting with subsurface water in the hyporheic zone. Thus the surface water receives more exchangeable Sr isotopes that have lower $^{87}Sr/^{86}Sr$ values from the sediments in the river bed and bank. This process can explain higher groundwater ratios in the stream using Sr isotopes during the non-irrigation period, compared with those estimated using water isotopes. This influence was relatively larger during the non-irrigation period because groundwater discharge was not sufficient, while during the irrigation period, the effects of WRI can be

negligible because of the higher rate in groundwater discharge from the regional aquifer.

In the previous manuscript, we concluded that Sr isotope is more stable in terms of space and time than water isotopes and has higher discriminating power for quantifying groundwater discharge compared with the water isotopes. However, in addition to this advantage, it should be noted that Sr isotopes can vary through WRI and might overestimate the groundwater contribution to streams, especially when the groundwater discharge does not dominate the hyporheic exchange rate. We would, therefore, conclude that the combined use of multiple tracers, including Sr and water isotopes and geochemical tracers, is recommended for examining the water mixture.

**[Changes in the manuscript]**

To reflect all these changes, we moved all the results regarding the endmember analysis from the discussion in the previous manuscript. In the revised manuscript, we also expanded the discussion, describing the use of multiple isotopes and geochemical tracer for understanding the hydrological cycle.

**Specific comments:**

*Almost all the references are cited in the introduction. The main text, and especially the discussion, should refer to appropriate references. Note that only 23 references are cited, which is not enough considering the 3 applied approaches and the abundant available literature available for each approach.*

**[Response]**

As described in the response to the general comment, we have expanded the discussion section on the contribution of groundwater discharge and hyporheic exchange from multiple tracers. We have also added several references accordingly.

*Samples dedicated to Sr isotopes analysis must be filtered and acidified to pH 2 with suprapure HNO3. Authors probably do it this way and this should be specified. Sr isotopes analysis: change 87Sr/86Sr of 8.37. . .. to 88Sr/86Sr (page 4, line 17).*

**[Response]**

The description has been corrected as follows.

**[Changes in the manuscript]**

*(previous manuscript)* we collected two bottles of water (50 mL), one for Sr isotopes and the other for stable isotopes of water. Both samples were filtered through 0.20 μm membrane filters.

*(revised manuscript)* we collected two 50-mL polyethylene bottles of water, one for Sr isotopes and the other for stable isotopes of water. Both samples were filtered through a disposable cellulose acetate filter (pore size, 0.2 μm; DISMIC 25CS020A5, Advantec, Tokyo, Japan).

*(previous manuscript)* The values of $^{87}Sr/^{86}Sr$ were normalized to a $^{87}Sr/^{86}Sr$ of 8.375209.
*(revised manuscript)* The values of $^{87}Sr/^{86}Sr$ were normalized to a $^{88}Sr/^{86}Sr$ of 8.375209.

*Sr concentration analysis method is not reported: please add it with the uncertainty.*
*Water isotopes analysis too poorly explained: especially using laser method, the salinity effect of the samples should carefully considered. Here we have no idea of the electric conductivity or TDS of the samples to evaluate a potential impact. More generally, the manuscript would benefit to have a brief overview of the major elements concentration.*

**[Response]**
Sr concentration and other rare elements were analyzed with ICP-MS, and the major elements were analyzed with ion chromatography. While we are currently working on the analysis of the principal component among these elements, which will be published in our next paper, we will add a table that describes the concentrations of the major and rare elements and water and Sr isotope ratios in the supplement of the revised paper.
The chloride concentration in the groundwater samples ranged from 2.83 to 13.18 mg/l during the irrigation period, and from 0.99 to 15.13 mg/l during the non-irrigation period. This low chloride concentration indicated that there have been little effects of salinity on the analysis water isotopes.

*For water isotopes, the uncertainty for both O and H measurements should be added.*
**[Response]**
We added the measurement errors for O and H in the revised manuscript. The relative errors of the standard material during the analysis for all the samples were less than 0.02‰ for $\delta^{18}O$ and 0.15‰ for $\delta^2H$.

*Section 2.2.1: specify the measurements period, over a same day, week , . . . ? what as the weather during that period ?*
**[Response]**
We have added a description about the weather and the measurement period. We also added a figure that depicts the seasonal variations in precipitation and groundwater table of the watershed.
**[Changes in the manuscript]**

The groundwater and surface water sampling were conducted during three consecutive days in each of the irrigation and non-irrigation period (21–23 June 2016 and 12–14 October 2016). To minimize the effect of precipitation on surface water sampling, we determined the sampling date in which cumulative precipitation for one week before the sampling periods were less than 20 mm. The average temperature during the surveys were 22 and 15°C for the irrigation and non-irrigation periods, respectively.

[Figure]

Figure 2 Seasonal variations in precipitation and depth of the groundwater table.

*Section 2.2.2: Equation 3 is not very clear, why not simply using for [Sr] and Sr isotopes (with R = 87Sr/86Sr, C = Sr concentration, f = fraction of endmember 1) : Rm = [f(R1\*C1)+ (1-f)(R2\*C2)] / Cm and Cm = f C1 + (1-f) C2 Specify that equation 4 is only true for stable isotopes of the water molecule.*

**[Response]**

Equation (3) was directly derived from the reference (Faure and Mensing, 2009), and it was used for explaining 'the 87Sr/86Sr of a mixture of two water samples A and B is a linear function of the inverse of the Sr concentration in the mixture'. For the estimation of the mixing ratio of water A and B, equation (4) in the previous manuscript can only be used for water isotope. We added the equation for the mixing ratio using Sr isotopes: $f_a(Sr) = (C_mR_m - C_bR_b) / (C_aR_a - C_bR_b)$. This equation is also derived from Faure and Mensing (2009).

**[Changes in the manuscript]**

The existing ratio of water sample A in the mixture, $f_a$, can be calculated using two isotopes: $^{87}$Sr/$^{86}$Sr and $\delta^{18}$O. The existing ratio of A estimated with $^{87}$Sr/$^{86}$Sr, $f_a$(Sr), can be calculated with Eq. (4):

$$f_a(Sr) = (C_m R_m - C_b R_b) / (C_a R_a - C_b R_b) \qquad (4)$$

The existing ratio estimated with $\delta^{18}$O, $f_a$(O), can be calculated with Eq. (5):

$$f_a(O) = (R_m - R_b) / (R_a - R_b) \qquad (5)$$

where $R_a$, $R_b$ and $R_m$ are the values of $\delta^{18}$O in the sample A, B and the mixture.

*Section 3.1.2: line 9: . . . because of mixing with fractionated water . . . Could we also consider that water has undergone direct evaporation and not only a mixing with an "older" evaporated water?*

**[Response]**

The concentration of Sr in the paddy with slower irrigation rate was 1.6 times higher than that with the higher rate, suggesting that 38% of ponded water was evaporated in the slower rate paddy. Changes in $\delta^{18}$O due to evaporation can be estimated with the Craig-Gordon model; and the model indicates $\delta^{18}$O increase by 3‰ from 38% of evaporation from the water surface. This is consistent with the difference in $\delta^{18}$O between the paddies with slower and faster irrigation rate, 3.8‰. Thus, we would argue that the increase in $\delta^{18}$O and $\delta^2$H in both slower rate paddy was simply owing to the evaporation from the water surface, not owing to the mixing with the fractionated water in the paddy.

**[Changes in the manuscript]**

*(previous manuscript)* water at the lower rate outlet was depleted in light isotopes, likely because of mixing with fractionated water in the paddy.

*(revised manuscript)* water at the lower rate outlet was depleted in light isotopes. Considering the Sr isotope result mentioned above, this result is likely due isotope fractionation by the evaporation of paddy water, which is derived from irrigation water and rain water.

*Section 3.2.1: in this section, the choice of the groundwater end-member (ODK / SAK) need to be better explained and argued as we note that the local wells present great variations for Sr isotopic signatures (figure 6).*

*In the same way in figure 7 and the text referring to this figure, the end-members "surface water" derived from the irrigation channel and the "groundwater end-member(s)" derived from the springs SAK and ODK should be explained.*

**[Response]**

As described in the response to the general comment, we created a new section (3.3.1) to explain how we determined the endmembers. Figures 6 in the previous manuscript that illustrated variations in Sr isotopic compositions of all the samples helps to highlight the groundwater sampled near the stream (within 200 m of the stream; red circles) is concentrated near the samples from the spring, ODK, while other sampling wells were highly scattered in the figure. We highlighted that the points presumably interact with the stream in a box with orange line.

We determined the endmember of the groundwater at the crossing of the two lines in Figure 7 (previous manuscript): the regression line of the stream water samples and the horizontal line from the plot of ODK ($^{87}Sr/^{86}Sr$ = 0.7074). For the surface water endmember, we also used the regression lines. The regression lines for the irrigation and non-irrigation periods crossed near the plot for the irrigation water; and we chose this crossing as the surface water endmember.

It is also worth noting that $^{87}Sr/^{86}Sr$ of the two springs in the upstream (SAK) and in the downstream (ODK) exhibited similar values, whereas $\delta^{18}O$ of them were totally different.

*Section 3.2.2:*

*lines 25-30: data plotting below the LMWL have an explanation, the studies cited give this explanation, it's not only an observation.*

**[Response]**

Thank you for bringing this to our attention. The cited references indicate that seasonal variations exist in deuterium excess (d-excess) in precipitation over large part of the East Asia. We have revised the expression accordingly.

**[Changes in the manuscript]**

*(previous manuscript)* … was consistent with the observation that …

*(revised manuscript)* … was consistent with the explanation that …

*Line13 page 9: data from the paddy define a line with a slope of 5.1, is this value in agreement with local annual humidity?*

**[Response]**

The relative humidity in this region is approximately 75% in summer (from June through to August) at the Utsunomiya meteorological station, located 20 km from the watershed. The

observed slope for the rice paddy water in this study, 5.1, was less steep than the estimated slope from the Craig-Gordon model (approximately 6) but consistent with the observed slope (ranging from 4 to 6) in other rice paddies in Japan (Hamada et al., 2004; Tsuchihara et al., 2011, Tsuchihara et al., 2016).

 **[Changes in the manuscript]**

*(previous manuscript)* the measured $\delta^{18}$O at the outlet suggests that the effects of kinetic fractionation resulted in different $\delta^{18}$O values that depended on the rate of irrigation

*(revised manuscript)* the measured $\delta^{18}$O and $\delta^2$H at the outlet suggests that the effects of kinetic fractionation mainly due to evaporation resulted in different $\delta^{18}$O and $\delta^2$H values that depended on the rate of irrigation.

> Tsuchihara, T., Yoshimoto, S., Shirahata, K., and Ishida, S. (2016): $^{17}$O-excess and stable isotope compositions of rainwater, surface water and groundwater in paddy areas in Ibaraki, Japan, Transactions of The Japanese Society of Irrigation, Drainage and Rural Engineering, 84 (2), I_185-I_194.
>
> Tsuchihara, T., Yoshimoto, S., Ishida S., and Imaizumi, M. (2011): Classification of recharge sources of groundwater in a paddy dominant alluvial fan based on geochemical and isotopic analyses, Technical report of the National Institute for Rural Engineering, 211, 21-34.
>
> Hamada, Y., Yabusaki, S., Tase, N., and Taniyama, I. (2004): Stable isotope ratios of Hydrogen and Oxygen in paddy water affected by evaporation, Journal of Japanese Association of Hydrological Sciences, 34(4), 209-216.

*Section 4.1:*

*line 17 : specify why percolation has no effect on sable isotopes of water molecule.*

*Lines 23-24: not clear, do you mean that spring (ODK) water reflect the buffering effect of the aquifer?*

**[Response]**

Gehrels et al. (1998) observed $\delta^{18}$O in soil water at different depths and found that the values near the ground surface varied in time, reflecting temporal variations in recharged water, while it converged with depth to the average of the variation. Water isotopes obtained at depths of 1.0 and 1.5 m were almost similar and close to the values of the average of two water samples obtained

in the paddy fields with different irrigation rates. This observation suggests that the water isotopes in the soil can average the variations in the isotopic compositions from rice paddies.

The values of the water isotopes obtained at the soil water sampling plot, which is close to the apex of the fan, were similar with the values obtained at the spring, ODK, which is located at the toe of the fan. This can be explained by the buffering effect of the aquifer.

**[Changes in the manuscript]**

*(previous manuscript)* While the stable isotopes in ponded water changed in many ways, percolation appeared to have little effect on water isotopes in subsurface flow, …

*(revised manuscript)* While the stable isotopes in ponded water changed in many ways, the isotopic composition appeared to converge to the average of the surface water variation through percolation.

*(previous manuscript)* This lower variability suggests that spring water isotopes were spatial and temporal averages.

*(revised manuscript)* This lower variability suggests that spring water isotopes reflected the buffering effects of the regional aquifer.

> Gehrels, J.C., Peeters, J.E.M., De Vries, J.J., Dekkers, M. (1998) The mechanism of soil water movement as inferred from $^{18}$O stable isotope studies, Hydrological Sciences Journal, 43(4), 579-594.

*Section 4.2.1:*

*lines 6 to 16: this part can be shortened (too descriptive in the discussion section).*

*Figure 12 can also be removed.*

**[Response]**

As described above, we would like to retain these figures because they support our argument.

*Line 31 and followings: How is calculated the fraction of groundwater estimated from Sr isotopes in Br 1 and Br 23? from figure 7, Br1 is constituted of less than 10% of groundwater and Br23 has the signature of the groundwater end-member, i.e. almost 100% of groundwater . . . Please clarify.*

*Same thing for the global calculations in figure 13 and 14. In figure 14, colors are inversed compared to figure 13, to be checked.*

**[Response]**

As described in the response to the general comment, we have clarified the endmember determination and corrected the inconsistency in the calculated groundwater ratios in Figures 13 and 14 (previous manuscript). The groundwater ratios at Br-23 constituted more than 90% of groundwater for both the irrigation and non-irrigation periods. We examined the similarity and discrepancies of the estimated groundwater ratios using Sr and water isotopes and discussed it, incorporating the chloride concentration of the stream. We also corrected the use of colors in these figures (see above response to general comment).

---

## Author Comment (AC2) · 19 Feb 2019

Reply to comments from Anonymous Referee #2.

**General Comments**

*The manuscript attempts to quantify seasonal variabilities in groundwater discharge in an extensive irrigation watershed using H, O, and Sr isotopes. conducted point- and watershed-scale observations of surface water, soil water, groundwater, and ponded water in rice paddies and examined changes in these isotopic compositions. They conclude that the ratios of groundwater to the stream was in the range 7–86% during the irrigation period and 38– 66% during the non-irrigation period. The use of Sr isotopes showed better results that those of stable water isotopes. The manuscript contains some useful material, however in its current form the manuscript is not publishable. It seems particularly apparent given the amount of time the manuscript spends describing the measurement results itself. While no quantitatively evidence to support their assumptions, e.g. the water isotope diagram can not provide direct evidence. And despite the length there are several statements in the technical description which need to be clarified as they indicate some further analysis is required to confirm the results.*

**[Response]**

We thank the reviewer for his/her time towards reading our manuscript and providing thoughtful comments. Our responses to the comments are as follows.

**Major comments:**

*1. I don't find the new insight from this paper. Since Sr isotopes is less fractionation, it is well known that the use of Sr isotopes has the higher potential to aid in quantification of temporal variations in groundwater discharge.*

**[Response]**

In a previous manuscript, we concluded that Sr isotope is more stable in terms of space and time than water isotopes and has higher discriminating power for quantifying groundwater discharge. As the reviewer suggested, this is consistent with previous findings. However, to our knowledge, studies that aimed to quantify groundwater discharge using the stability of Sr isotopes in groundwater have been quite rare. The groundwater table of the study watershed drastically changes between irrigation and non-irrigation periods (see Figure 2 below), thus the groundwater discharge from the regional aquifer significantly differs. This is the first study that illustrated the seasonal variation of groundwater discharge using Sr isotopes. We also pointed out it should be

noted that Sr isotopes may vary through water-rock interactions and might overestimate the groundwater contribution to streams, especially when groundwater discharge does not dominate the hyporheic exchange rate. We updated sections 4.1 and 4.2 to verify the robustness of the endmembers and the consistency of estimated groundwater ratios in the stream using two isotopes. We also added chloride concentration to corroborate our conclusion. We would, therefore, conclude that the combined use of multiple tracers, including Sr and water isotopes and geochemical tracers, is recommended for examining the water mixture.

**[Changes in the manuscript]**

To reflect all these changes, we moved all the results regarding the endmember analysis from the discussion in the previous manuscript. In the revised manuscript, we also expanded the discussion, describing the use of multiple isotopes and geochemical tracer for understanding the hydrological cycle.

*2. The manuscript is not straightforward, and the results are difficult to understand. For me, it is better to show the sampling locations in more detailed way. I can not find where is the location of Br1- 23. And where is the upstream and downstream mentioned in the manuscript. Also, please give the detail information about sampling date. This is extremely important for stable water isotope study. If surface water, soil water, groundwater, and ponded water in rice paddies are sampling in different days, the authors should make sure they are not change significantly in temporal scale.*

**[Response]**

The location and date of sampling are now specified in the revised manuscript. To understand the relationship among the sampling locations, we integrated Figures 1, 2 and 3 into one figure. For Figures 2 and 3 in the previous manuscript, we added the location of the bridges along the Gogyo River (sampling location of stream water) and the upstream and downstream.

The sampling was conducted within 3 consecutive days of the survey, during the period no precipitation was recorded. In addition, the effect of precipitation is considered to be negligible because cumulative precipitation before the survey was less than 20 mm.

As the reviewer pointed out, the stable isotopes of water show temporal variation, and the degrees of the variations are the largest in precipitation, followed by ponded water in rice paddies, soil water, and groundwater. The point-scale survey of this study confirmed that the variation of soil water and groundwater were less than those of ponded water in rice paddies. The scope of this study is to examine the interactions between streamflow and groundwater, rather than

investigating changes in the stable water isotopes through irrigation, evaporation and percolation in rice paddies. For this objective, we showed the seasonal variations of stable water isotopes in groundwater and identified suitable endmembers that have the least seasonal variations. Indeed, the temporal variations in stable water isotopes may induce uncertainties in the estimated groundwater discharge and we will address this issue in our ongoing research.

**[Changes in the manuscript]**

[Figure]

Figure 1 Overview of the study watershed: (a) Kinu river watershed, (b) sampling location in the Gogyo river watershed. Shaded area is depicted in (c), (c)

The groundwater and surface water sampling were conducted during three consecutive days in each of the irrigation and non-irrigation periods (21–23 June 2016 and 12–14 October 2016). To minimize the effect of precipitation on surface water sampling, we determined the sampling date in which cumulative precipitation for one week before the sampling periods were less than 20 mm. The average temperature during the surveys were 22 and 15°C for the irrigation and non-irrigation periods, respectively.

*3. Ponded water isotope in rice paddies indeed showed large spatial variability. The inflow side and outflow side will show large differences. I don't know whether the authors consider this or not. To get an average value, I think it need special treatment.*

**[Response]**

As we replied to the previous comment, numerable studies showed that the temporal variation in stable water isotopes was large. In addition to the temporal variation in precipitation, the effects of kinetic fractionation in rice paddies would increase the variabilities.

Gehrels et al. (1998) observed $\delta^{18}$O in soil water at different depths and found that the values near the ground surface varied in time, reflecting temporal variations in recharged water, while it converged with depth to the average of the variation. From the point-scale survey, we showed that the water isotopes obtained at the depth of 1.0 and 1.5 m were almost similar and were close to the values of the average of two water samples obtained in the paddy fields with different irrigation rates. Our observation was consistent with the findings of Gehrels et al. (1998) and suggested that the water isotopes in the soil can average the variations in the isotopic compositions from rice paddies.

The values of the water isotopes obtained at the soil water sampling plot, which is close to the apex of the fan, were similar with the values obtained at the spring, ODK, which is located at the toe of the fan. This can be explained by the buffering effect of the aquifer.

**[Changes in the manuscript]**

*(previous manuscript)* While the stable isotopes in ponded water changed in many ways, percolation appeared to have little effect on water isotopes in subsurface flow, …

*(revised manuscript)* While the stable isotopes in ponded water changed in many ways, the isotopic composition appeared to converge to the average of the surface water variation through percolation.

*(previous manuscript)* This lower variability suggests that spring water isotopes were spatial and temporal averages.

*(revised manuscript)* This lower variability suggests that spring water isotopes reflected the buffering effects of the regional aquifer.

Gehrels, J.C., Peeters, J.E.M., De Vries, J.J., Dekkers, M. (1998) The mechanism of soil water movement as inferred from $^{18}$O stable isotope studies, Hydrological Sciences Journal, 43(4), 579-594.

*4. The most important thing is neglecting the effect of precipitation. Please plot out the precipitation during the sampling period. Precipitation will definitely change all the results.*

**[Response]**

As described in the response to the first comment, we added a description of the weather conditions during the sampling period, and it was suggested that the direct effects of precipitation on the surface water samples were negligible. The water isotope of the precipitation was plotted in Figure 4(a). The red dashed line in Figure 4(b) indicates the direction for the precipitation plot. We added an inset to Figure 4(b) that shows all the sample plots including the precipitation.

The precipitation apparently affected surface water (e.g., Sr isotopic compositions of water in the rice paddies through dilution); however, the temporal variations in precipitation would not change the values of Sr and water isotopic compositions in the groundwater because of large buffering effect of the aquifer. This study aims to examine the relationship between streamflow and groundwater. Thus, we would argue that this comment from the reviewer missed the point we are trying to make.

**[Changes in the manuscript]**

[Figure]

Figure 4 Water isotope diagram and $^{87}$Sr/$^{86}$Sr–1/Sr diagram for the point-scale survey.

*5. The two endmembers partitioning method is good but neglecting the recharge process. This may be important for irrigation period. Large portions of irrigation water may recharge regional aquifer. Any idea or evidence?*

**[Response]**

This study focuses on how the recharge from rice paddies may or may not affect the isotopic compositions of regional aquifers. To address this point, we conducted the point-scale survey and examined how Sr and water isotopic compositions change through percolation. We found that Sr isotopes change relatively within a short time through water-rock interactions and reached an equilibrium to the Sr isotopic composition of the lithology. On the other hand, the water isotopes in the recharged water was significantly affected by evaporation at rice paddies and change the groundwater isotopic compositions. We would argue that the recharge processes were not neglected, but they are rather included as the central topic of this study.

**[Changes in the manuscript]**

Added an explanation in the 'Study watershed' (2.1.1) and a figure depicting the seasonal variations in precipitation and depth to the groundwater table (see Figure 2, response to the following comment).

To illustrate an overview of water balance, we added a figure depicting seasonal variations in precipitation and groundwater table of the watershed. The figure includes the duration of irrigation for rice paddies (from April 15 through August 31) to indicate that the regional aquifer receives a substantial volume of recharge from ponded rice paddies and the groundwater table increases by about 2 m during the irrigation period.

Indeed, large portions of irrigation water recharge the regional aquifer and increase the regional water table by approximately 2 m. Consequently, the regional aquifer becomes the major source of groundwater discharge during irrigation seasons.

*6. To give some quantitative conclusions from the isotope experiment, I suggest the authors should at least discuss the result with water balance components (precipitation, ET, irrigation, flow rates in rivers, groundwater table fluctuations, etc.) at the specific studied area. For instance, the authors can estimate the irrigation water based on the local irrigation schedule and the cultivated area. Currently, the authors only present the peak flow rates for the whole diversion weirs (71 m^3/s), it is hard for us to link this to your experimental results.*

**[Response]**

Along the reviewer's comment, we have added the following explanation to illustrate the characteristic hydrological cycles of the study watershed (Figure 2). As depicted in Fig. 2, the groundwater table is raised by about 2 m at the onset of the irrigation (early May).

The streamflow during the irrigation period fluctuated as depicted as 'Observed streamflow' in Figure 10 and 11 in the previous manuscript. As shown in Fig 10 (see below) and the description in the previous manuscript, it is very difficult to grasp the water balance from the streamflow data alone because there were substantial number of inflow and outflow channels to the stream: '*We measured the rates of lateral inflow at 30 channels (drainage from surrounding rice paddies) and of outflows at 22 channels (diversions from stream to paddies)*'.

Because these two points would be important background why we conducted this study, we added descriptions to clarify them.

**[Changes in the manuscript]**

[Figure]

Figure 2 Seasonal variations in precipitation and depth of the groundwater table.

[Figure]

Figure 10 Estimated groundwater discharge to stream based on the observed water balance during irrigation period.

**Minor comments:**

*Line 24 p5: The – the*

The error has been corrected.

*Line 25 p6: water table was 1.67 m — this is confusing. Do you mean groundwater depth?*

Thank you for pointing this out. The depth has been specified: the groundwater depth was 1.67 m from the ground surface.

*Line 21 p9: Usually observation error is not portable, given the data features at different basins are quite different. Moreover, the reference you cite is from 1963, now we have more accurate and convenient method to measure the flow rate.*

**[Response]**

There are several other papers that show observation errors in stream flow measurement. For example, McMillan et al. (2012) summarized uncertainties from the extensive review of the streamflow measurements; and the typical errors in streamflow measurement are 50–100% for low flows, and 10–20% for medium to high flows. Hence, we would argue that our assumption for the errors in stream flow measurement of 15% is reasonable.

**[Changes in the manuscript]**

McMillan et al. (2012) summarized uncertainties through an extensive review of the streamflow measurements, and typical errors in streamflow measurement are 50–100% for low flows, and 10–20% for medium to high flows.

*Lines 6, 8 in p11: leave a space after ‰*

A space has been inserted.

*Line 7 p12: duplicated ,*

The duplication has been eliminated.

*Line 21 p13: The – the*

The error has been corrected.

*Line 26 p13: please use endmember or end member consistently in the paper.*

We have ensured consistent use of 'endmember' the revised manuscript.

---

## Author Comment (AC3) · 19 Feb 2019

The comment was uploaded in the form of a supplement.

Please also note the supplement to this comment:
https://www.hydrol-earth-syst-sci-discuss.net/hess-2018-551/hess-2018-551-AC3-supplement.pdf